# XIL: CROSS-EXPANDING INCREMENTAL LEARNING

**Heayoun Choi, Hyundong Jin, Eunwoo Kim***
Chung-Ang University
heayounchoi@cau.ac.kr, wlsgusehd@gmail.com, eunwoo@cau.ac.kr

## ABSTRACT

Class-Incremental Learning (CIL) traditionally assumes that all tasks share a similar domain distribution, limiting its applicability in real-world scenarios where data arrive from evolving environments. We introduce a new problem setting, *Cross-Expanding Incremental Learning (XIL)*, which extends CIL by requiring models to handle class-incremental data across distinct domains and to expand class-domain associations bidirectionally. In this setting, new classes should be integrated into previously seen domains, while earlier classes are extended to newly encountered ones, a capability we refer to as *bidirectional domain transferability (BiDoT)*. To address XIL, we present a new framework, *Semantic Expansion through Evolving Domains (XEED)*, which leverages domain-specialized prompts, residual-guided representation modulation, and evolving prototype embeddings to expand class semantics across previously encountered domains. We further introduce the *BiDoT Score*, a novel metric for quantifying the degree of BiDoT. Extensive experiments on benchmark datasets with significant domain shifts demonstrate that XEED outperforms existing CIL baselines by a large margin in both standard accuracy and BiDoT scores, establishing a strong foundation for realistic continual learning under domain-evolving conditions.

## 1 INTRODUCTION

Class-incremental learning (CIL) aims to enable models to incrementally learn new classes while retaining previously acquired knowledge (Zhou et al., 2024b). To address this challenge, a wide range of learning strategies have been proposed (Li & Hoiem, 2017; Rebuffi et al., 2017; Kirkpatrick et al., 2017; Zenke et al., 2017; Aljundi et al., 2018; Yan et al., 2021; Douillard et al., 2022), and prompt tuning approaches (Wang et al., 2022d;c; Smith et al., 2023; Gao et al., 2024; Qu et al., 2025) have attracted increasing attention for their ability to leverage pre-trained models and adapt them to new tasks with minimal computational overhead, often surpassing full fine-tuning in performance.

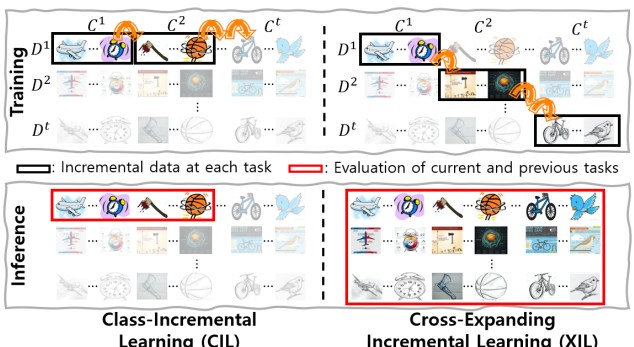

Figure 1: Comparison of training and inference phases between conventional class-incremental learning (CIL) and our proposed cross-expanding incremental learning (XIL). While CIL assumes a shared domain across all incremental tasks, XIL considers distinct domains and requires bidirectional semantic expansion, extending new classes to past domains and past classes to new ones.

However, most CIL methods rely on the assumption that the training and test data share a similar domain distribution, making them susceptible to performance degradation under domain shifts (Kundu et al., 2020). While a number of approaches (Shi & Wang, 2023; Kundu et al., 2020; Simon et al., 2022; Cho et al., 2023; Kishida et al., 2021) have been introduced to alleviate the domain gap mismatch between incremental tasks by incorporating perspectives from domain adaptation (Ganin & Lempitsky, 2015) and domain generalization (Zhou et al., 2022), they remain limited in scope. *In this paper, we systematically compare existing tasks and propose a novel task that has not yet been explored in the literature, together with a simple baseline framework.*

---

*Corresponding author

Figure 2: Gap between the accuracy after the final task in XIL and in CIL for full fine-tuning (Joint-FT), recent state-of-the-art methods, and the proposed framework (XEED) on datasets with substantial domain shifts. A more negative gap indicates poorer generalization across domains.

Problem settings tackling the domain gap mismatch in incremental scenarios generally assume a fully or partially shared label space across domains, enabling the transfer of attributes common to classes. Shi & Wang (2023) consider a setting where the label space remains fixed across tasks, requiring the integration of newly encountered domain knowledge over time in a supervised manner. Kundu et al. (2020) study a semi-supervised scenario in which a model must adapt to a test domain with few labeled examples, while De Carvalho et al. (2024); Weng et al. (2024) consider settings that rely on unlabeled target-domain examples. Simon et al. (2022) investigate a multi-domain training setup where the goal is to learn invariant features that generalize to unseen test domains. Similarly, Cho et al. (2023) tackle the joint challenge of domain adaptation and domain generalization to unseen distributions. Moreover, Kishida et al. (2021) investigate scenarios with unknown classes, in which the model is tasked with detecting novel objects and simultaneously adapting to known classes across domains.

In practice, data availability varies significantly across domains and classes, and there may be no data at all for learning shared attributes of classes across domains. For example, consider an industrial robot trained to recognize components using high-quality images collected in a controlled factory environment. When deployed in different facilities or adapted to new tasks, the robot may need to learn additional parts from operator-provided data, such as smartphone photos, technical diagrams, or even hand-drawn sketches. In such cases, previously learned components may not appear in the new environment at all. Moreover, the robot's working environment can change frequently through relocation or reverting to previous settings, requiring it to recognize previously learned components across all encountered domains, a capability that lies beyond the scope of conventional CIL settings.

Therefore, we introduce a novel problem setting, *Cross-Expanding Incremental Learning (XIL)*. As illustrated in Figure 1, XIL requires a model not only to retain previously learned knowledge as new classes emerge from different domains over time, but also to bidirectionally transfer domain knowledge across learned classes. It should generalize to previously encountered domains where direct supervision was unavailable for certain classes. We refer to this capability as *bidirectional domain transferability (BiDoT)*, and to evaluate it, we introduce a new metric, *BiDoT Score*.

The central question we raise is whether current models are capable of transferring domain knowledge across classes when class distributions are not shared across domains. Using the accuracy gap between XIL and CIL, we empirically show that neither full fine-tuning nor recent state-of-the-art methods (Wang et al., 2022d;c; Smith et al., 2023; Gao et al., 2024) exhibit inherent bidirectional domain transferability. As shown in Figure 2, performance drops significantly when the model is faced with unseen class-domain combinations, indicating that such generalization is not naturally supported by current architectures or training protocols. To address this, we propose a new framework, *Semantic Expansion through Evolving Domains (XEED)*, inspired by generative replay-based methods using synthetic images. XEED leverages generative models to transfer domain characteristics to class-specific attributes by exploiting semantic relationships between classes and domain-specific features. Specifically, XEED first learns domain-specialized prompts via an auxiliary classifier. It then employs prototype embeddings to progressively expand the semantic representation space, enabling bidirectional integration of domain knowledge from synthetic samples.

Since the XIL task is novel and no directly comparable baselines exist, we conduct experiments using the strongest and most relevant prompt-based learning methods, selected after a comprehensive consideration of CIL variations (Zhou et al., 2023; Kundu et al., 2020; Simon et al., 2022; Cho et al., 2023; Kishida et al., 2021), generative replay-based approaches (Meng et al., 2024; Zhang et al., 2025; Wu et al., 2025), and prompt-based techniques (Wang et al., 2022d;c; Smith et al., 2023; Gao et al., 2024). Extensive experiments on datasets with substantial domain shifts demonstrate that XEED consistently outperforms existing CIL methods in both standard accuracy and BiDoT Scores.

## 1.1 CONTRIBUTIONS

Our key contributions are as follows: (1) We propose a novel problem setting, *Cross-Expanding Incremental Learning (XIL)*, which, to the best of our knowledge, is the first to extend class-incremental learning to enable bidirectional domain knowledge transfer across previously learned classes. (2) We propose a new framework, *Semantic Expansion through Evolving Domains (XEED)*, which integrates newly introduced knowledge and enables bidirectional domain transfer by leveraging semantic relationships between classes and domains. (3) We introduce a new evaluation metric, the *BiDoT Score*, to measure the *bidirectional domain transferability (BiDoT)*, an aspect not addressed in CIL. (4) We conduct extensive experiments and analyze the limitations of existing CIL baselines under the XIL setting. XEED outperforms baselines with 7.1% higher average standard accuracy, and up to 31.41% higher BiDoT scores across all datasets.

## 2 RELATED WORKS

### 2.1 CLASS-INCREMENTAL LEARNING

Class-incremental learning (CIL) methodologies can be broadly categorized based on the learning strategies they employ. Strategies such as knowledge distillation, regularization, and network expansion aim to retain prior knowledge while learning new tasks by transferring past information (Li & Hoiem, 2017; Rebuffi et al., 2017), constraining important parameters (Kirkpatrick et al., 2017; Aljundi et al., 2018), or dynamically extending the model architecture (Yan et al., 2021; Douillard et al., 2022). Unlike such conventional approaches, prompt tuning methods leverage pre-trained models to adapt to downstream tasks with minimal resource consumption (Wang et al., 2022c; Smith et al., 2023; Gao et al., 2024; Qu et al., 2025), achieving competitive or even superior performance compared to full fine-tuning at a much lower cost.

Another common approach is to store a subset of training data as exemplars and replay them during new task learning to retain prior knowledge (Rebuffi et al., 2017; Hou et al., 2019; Wang et al., 2022a). However, growing concerns over privacy and regulations have led to a shift toward generating synthetic images that resemble previous classes, rather than storing real user data (Meng et al., 2024; Krawczyk & Gepperth, 2024; Wu et al., 2025; Zhang et al., 2025). Inspired by these developments, our framework leverages generative models to generate synthetic cross-class domain transferred images and learns domain-specialized prompts to adapt to diverse domains.

### 2.2 MORE PROBLEM SETTINGS IN CIL

Recent studies (Wang et al., 2023; Shi & Wang, 2023; Liu et al., 2024; Ganin & Lempitsky, 2015; Zhou et al., 2022) have challenged core assumptions in CIL, such as tasks sharing the same domain distribution or training and test data originating from the same domain. For settings where domain distributions vary across tasks with label space remains fixed, Liu et al. (2024) propose using batch-wise exponential moving average algorithm to adaptively mitigate classifier forgetting. Similarly, Shi & Wang (2023) introduce adaptive coefficient learning to dynamically adjust replay, cross-domain distillation, and divergence terms. Wang et al. (2023) consider significant domain gaps during training, similar to XIL, but their objective is to mitigate knowledge interference across tasks by isolating task-specific knowledge. Their approach does not handle scenarios where class-domain associations change at test time, requiring generalization across both domain shifts and class boundaries. In contrast, XIL focuses on bidirectional domain knowledge transfer across tasks.

Other works (Kundu et al., 2020; Simon et al., 2022; Cho et al., 2023; Kishida et al., 2021) combine CIL with domain adaptation or generalization. Kundu et al. (2020) leverage Gaussian prototype-based distribution information and guides to distinguish new classes under test-time domain shifts and adapt source-trained models. Simon et al. (2022) use multi-source domain data to learn domain-invariant features via class-wise Mahalanobis metrics, enabling generalization to unseen domains. Cho et al. (2023) propose a complementary framework in which a domain adaptation model provides pseudo-labels for domain generalization, while the domain generalization model initializes the domain adaptation model, thereby addressing the joint challenge of domain adaptation and generalization. These approaches rely on strong assumptions: Kundu et al. (2020) require labeled target data for both old and new classes, and Simon et al. (2022); Cho et al. (2023) assume access to diverse domains covering all classes. In contrast, XIL sees each class in only one source domain and must transfer knowledge bidirectionally to generalize across unseen domains.

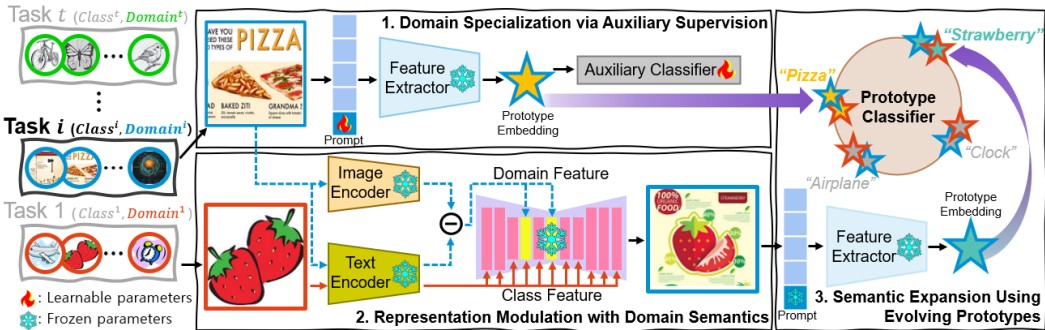

Figure 3: Overview of the XEED framework at task $i$. Prototype embedding fill colors represent classes, and outline/prompt colors represent domains. (1) During each task, domain-specialized prompts are concatenated with image patches and optimized to encode domain-specific semantics via auxiliary supervision. (2) A diffusion model generates synthetic images by disentangling and recombining class and domain semantics across tasks, enabling bidirectional domain transfer. (3) The classifier updates its class prototypes with generated data, leading to an evolving semantic space that captures richer class semantics across all seen domains. Best viewed in color.

## 3 PROBLEM STATEMENT

In *Class-Incremental Learning (CIL)* (Zhou et al., 2023), the label space $C \subset Y$, where $C$ represents a subset of classes from the complete class set $Y$, expands incrementally as new class subsets are introduced over time, while the domain distribution $D$ remains fixed. The training sequence is $T_{\text{CIL}} = \{(C^1, D), (C^2, D), \dots, (C^t, D)\}$, where $C^i \subset Y$, $C^i \cap C^j = \emptyset, \forall i \neq j$, and $t$ denotes the total number of incremental learning steps. During inference, the model is evaluated on all previously learned classes within the same domain encountered during training: $\mathcal{E}_{\text{CIL}} = \bigcup_{i=1}^{t}(C^i, D)$.

While CIL assumes a fixed domain distribution during incremental class learning, real-world scenarios often involve evolving domains. To bridge this gap, we introduce *Cross-Expanding Incremental Learning (XIL)*, a novel setting in which a model should incrementally learn new classes from evolving domains while retaining prior knowledge. In addition, this setting aims to enable bidirectional transfer of domain knowledge across classes, including to domains where certain classes were never directly supervised. Formally, the label space in XIL expands over time with the introduction of new class subsets as in CIL. However, the domain distribution $D$ also varies across tasks. Each domain $D$ comprises samples drawn from a unique distribution $S = \{(x_k, y_k)\}_{k=1}^{n} \sim P_{XY}$, where $x_k \in X$ is an input, $y_k \in C \subset Y$ is the corresponding label, $n$ is the number of samples, and $P_{XY}$ is the joint input-label distribution (Wang et al., 2022b). A domain shift between tasks is defined as $P_{XY}^i \neq P_{XY}^j$ for all $i \neq j$. XIL defines a sequence of tasks as: $T_{\text{XIL}} = \{(C^1, D^1), (C^2, D^2), \dots, (C^t, D^t)\}, C^i \subset Y, C^i \cap C^j = \emptyset, \forall i \neq j$.

During inference, the model is evaluated not only on previously observed class-domain pairs, but also on its ability to generalize to novel combinations, i.e., classes appearing from encountered but unsupervised domains. This evaluation is defined as: $\mathcal{E}_{\text{XIL}} = \bigcup_{i=1}^{t} \bigcup_{j=1}^{t}(C^i, D^j)$, where $D^j = D^i$ or $D^j \neq D^i$, capturing both knowledge retention and bidirectional domain transfer across tasks.

## 4 THE PROPOSED FRAMEWORK

Our framework, XEED (Semantic Expansion through Evolving Domains), tackles the challenge of bidirectional domain transferability in XIL by incrementally expanding class semantics across all previously encountered domains. It consists of three core components that work in concert to support semantic expansion across tasks, as illustrated in Figure 3. First, domain-specialized prompts are learned via auxiliary supervision to encode domain-specific characteristics while remaining disentangled from class semantics. Then, synthetic images are created by modulating class representations with domain features through residual-guided cross-attention. Finally, prototype embeddings are continuously updated with synthetic images, allowing the semantic representation space to evolve bidirectionally, extending newly learned classes to previously encountered domains and past classes to newly encountered domains.

### 4.1 OUR FRAMEWORK: SEMANTIC EXPANSION THROUGH EVOLVING DOMAINS (XEED)

#### 4.1.1 DOMAIN SPECIALIZATION VIA AUXILIARY SUPERVISION

In XIL, the domains of class-incremental data continuously change over time, and the model should be able to extract meaningful features for learned classes across all previously seen domains, even without direct supervision. To address this variability, we employ an auxiliary classifier during training to learn small domain-specialized prompts (Jia et al., 2022) that effectively capture underlying domain characteristics, facilitating adaptation across multiple domains.

Given a task $(C^t, D^t) \in T$ consisting of input-label pairs $(X^t, Y^t)$, where $C^t = Y^t \subset Y$ denotes the label subset at timestep $t$ and $D^t$ is the associated domain, we introduce a domain-specific prompt $P^{D^t}$ to inject specific domain information of the task into a frozen pre-trained feature extractor $f_\phi$. Each image sample $x_i \in X^t$ is first mapped to patch embeddings $x_{\text{img}} \in \mathbb{R}^{N \times D}$, where $N$ is the number of image patches, and $D$ is the embedding dimension. A class token $x_{\text{cls}} \in \mathbb{R}^{1 \times D}$ is prepended, and a learnable prompt $P^{D^t} \in \mathbb{R}^{L \times D}$ designed to encode domain-specific information, where $L$ is the length of the prompt token, is inserted after the class token, yielding the final input:

$$z_i = [x_{\text{cls}}, P^{D^t}, x_{\text{img}}] \in \mathbb{R}^{(1+L+N) \times D}. \tag{1}$$

The sequence $z_i$ is processed by the frozen feature extractor $f_\phi$, and the output at the class token position is used as the image-level representation:

$$h_i = f_\phi(z_i)[0]. \tag{2}$$

To ensure the prompt $P^{D^t}$ captures domain-level rather than class-specific features, an auxiliary linear classifier $A_\phi$ is introduced during training. It takes $h_i$ as input and is trained with a cross-entropy loss to predict the class within domain $D^t$. During training, only the prompt $P^{D^t}$ and the classifier $A_\phi$ are updated, while all parameters of the feature extractor $f_\phi$ remain frozen. The training objective of the current task dataset $\mathcal{S}_t = \{(x_i, y_i)\} \sim P^t_{XY}$ is defined as:

$$\mathcal{L}_{\text{CE}} = - \sum_{(x_i, y_i) \in \mathcal{S}_t} y_i \log A_\phi(h_i). \tag{3}$$

This auxiliary supervision acts as a regularizer, guiding the prompt to encode shared domain characteristics while avoiding class-specific semantics. As a result, the prompt effectively conditions the representation space of the feature extractor based on the domain $D^t$ of the input image.

#### 4.1.2 REPRESENTATION MODULATION WITH DOMAIN SEMANTICS

To address unsupervised domain gaps in rehearsal-free CIL, we leverage the generative capability of a pre-trained diffusion model conditioned on both image and text inputs (Ye et al., 2023). Specifically, we transfer domain semantics to class-specific attributes via representation modulation in a training-free manner. To enable bidirectional domain knowledge transfer, we extract class centroids from sequential data to generate and store pseudo-exemplars, discarding the original data.

Exemplar generation incorporates a sequentially incoming image $x_i$ as the image conditioning input $I_i$, and the class name corresponding to label $y_i$ as the text conditioning input $T_i$. Both inputs are encoded using a CLIP image encoder $\text{Enc}_I$ and a CLIP text encoder $\text{Enc}_T$, respectively. At task $t$, we define an exemplar $x_{\text{ex}}$ as:

$$x_{\text{ex}} = g_\theta(z, k, \text{Enc}_T(T_i^t), \text{Enc}_I(I_i^t)), \tag{4}$$

where $g_\theta$ is a diffusion model parameterized by $\theta$, $z$ is random noise, and $k$ denotes the total number of denoising steps. Using a small $k$ preserves high-level semantics while modifying low-level details, mitigating overfitting to the original image and enabling faster generation.

For bidirectional domain knowledge transfer from one task $t$ to another $t' \neq t$, we first suppress class-specific semantics in $I_i^t$ and retain only domain-related features. We compute a residual vector by subtracting the class embedding from the image embedding (Ye et al., 2023):

$$\delta_I^t = \text{Enc}_I(I_i^t) - \text{Enc}_T(T_i^t), \tag{5}$$

which is used as the image conditioning input for generation:

$$x_{\text{transfer}}^{t' \leftarrow t} = g_\theta(z, k, \text{Enc}_T(T_i^{t'}), \delta_I^t). \tag{6}$$

To preserve the class-specific semantics of $T_i^{t'}$ while injecting domain features from $\delta_I^t$, cross-attention (Vaswani et al., 2017) is modified to incorporate $\delta_I^t$ only in specific transformer blocks responsible for layout and style (Wang et al., 2024). This enables modulation of the representation only in terms of the domain semantics of $T_i^{t'}$.

### 4.1.3 SEMANTIC EXPANSION USING EVOLVING PROTOTYPES

To continuously expand the semantic representation space of the classifier using generated data, we employ evolving prototype embeddings that are incrementally updated as new domain knowledge arrives. Specifically, each class $c \in C$ within a domain $D^t \in D$ is represented by a domain-aware prototype vector $\mu_c^{D^t}$. This vector is computed as the mean of feature embeddings extracted by the frozen feature extractor $f_\phi$, conditioned on the corresponding domain-specialized prompt:

$$\mu_c^{D^t} = \frac{1}{|\mathcal{E}_c^{D^t}|} \sum_{x_i \in \mathcal{E}_c^{D^t}} f_\phi(x_i, P^{D^t}), \tag{7}$$

where $\mathcal{E}_c^{D^t}$ denotes the evolving support set for class $c$ in domain $D^t$. As new samples are synthesized or encountered, $\mathcal{E}_c^{D^t}$ is expanded accordingly, allowing $\mu_c^{D^t}$ to dynamically reflect the updated semantic structure of each class. Given a test input $x$, its feature embedding $h$ is classified by computing the cosine similarity with the current set of prototypes:

$$\hat{y} = \arg\max_{c \in C} \frac{\langle h, \mu_c^{D^t} \rangle}{\|h\| \|\mu_c^{D^t}\|}. \tag{8}$$

To select the appropriate domain prompt $\hat{D}$ during inference, the test image embedding $f_\phi(x)$ is compared with domain prototypes $\mu^{D^t}$. The prompt $\hat{D}$ is chosen by minimizing the distance: $\hat{D} = \arg\min_{D^t \in D} \left\| f_\phi(x) - \mu^{D^t} \right\|_2$. Each domain prototype $\mu^{D^t}$ is computed by averaging the image embeddings within each class under domain $D^t$, and then taking the mean across all classes:

$$\mu^{D^t} = \frac{1}{|C^{D^t}|} \sum_{c \in C^{D^t}} \left( \frac{1}{|\mathcal{E}_c^{D^t}|} \sum_{x_i \in \mathcal{E}_c^{D^t}} f_\phi(x_i) \right), \tag{9}$$

where $C^{D^t}$ is the set of classes in domain $D^t$.

## 5 EXPERIMENTS

### 5.1 EXPERIMENTAL SETUP

**Datasets.** To properly evaluate under the proposed XIL setting, each class must be observable in every domain so that tasks can introduce new class-domain combinations and unseen combinations can be assessed at test time. We used widely adopted datasets for domain adaptation (Chang et al., 2023; Dayal et al., 2025) and generalization (Jiao et al., 2025; Jin et al., 2021). PACS (Li et al., 2017) shows large style variations across domains, while DomainNet (Peng et al., 2019) has strong domain style differences and severe label shifts. Office-31 (Saenko et al., 2010) reflects environmental differences with small within-domain variation. Classes were randomly shuffled and assigned to tasks by first distributing an equal share via integer division. The remaining classes were then sequentially allocated to tasks in task order. In PACS, leftover classes were assigned to domains other than Photo, where no prompt tuning was applied, to prevent overfitting to a single class.

**Evaluation Metrics.** We evaluate using continual learning accuracy metrics (Rebuffi et al., 2017; Li & Hoiem, 2017) and propose the *BiDoT Score* to measure bidirectional domain transferability in XIL. The accuracy for task $i$ is defined as: $ACC^i = \frac{1}{|\mathcal{E}^{1:i}|} \sum_{(x,y) \in \mathcal{E}^{1:i}} \mathbf{1}(\hat{y} = y)$, where $\mathcal{E}^{1:i}$

Table 1: Final and average performance metrics (BiDoT, accuracy) for each dataset, averaged over three runs. Best results are highlighted in bold, and second-best results are underlined.

| Method | PACS | | | | Office-31 | | | | DomainNet | | | |
|---|---|---|---|---|---|---|---|---|---|---|---|---|
| | F-BiDoT | A-BiDoT | Final | Avg | F-BiDoT | A-BiDoT | Final | Avg | F-BiDoT | A-BiDoT | Final | Avg |
| Joint-FT | 26.73 | 26.73 | 49.74 | 49.74 | 48.60 | 48.60 | 64.99 | 64.99 | 23.20 | 23.20 | 33.76 | 33.76 |
| LwF | 26.86 | 43.18 | 32.72 | 61.17 | 26.67 | 21.56 | 42.72 | 62.55 | 10.28 | 8.48 | 15.82 | 32.15 |
| EWC | 24.25 | 35.52 | 35.27 | 57.20 | 47.02 | 68.41 | 49.57 | 56.58 | 12.49 | 12.96 | 12.81 | 32.59 |
| S-Prompts | 18.85 | 26.90 | 43.48 | 65.91 | 35.60 | 25.00 | 54.81 | 67.33 | 5.24 | 4.92 | 15.05 | 35.69 |
| CODA-P | 24.68 | 46.73 | 37.43 | 69.60 | 67.39 | 58.13 | 75.84 | 80.27 | 29.41 | 25.49 | 36.08 | 49.68 |
| CPrompt | 33.78 | 43.18 | 43.02 | 66.86 | 63.37 | 53.58 | 73.90 | 79.02 | 29.71 | 24.81 | 37.26 | 50.60 |
| SimpleCIL | 26.56 | 42.89 | 40.51 | 69.01 | 69.06 | 60.93 | 75.45 | 77.12 | 11.94 | 11.10 | 17.00 | 32.25 |
| **XEED** | **65.19** | **67.77** | **61.86** | **78.51** | **78.08** | **73.67** | **80.72** | **83.19** | **33.63** | **37.57** | **35.30** | **48.40** |

contains all samples from tasks 1 to $i$, and $\mathbf{1}(\cdot)$ is the indicator function. We report average (*Avg*) and final (*Final*) accuracy after all tasks.

In addition, we introduce the *BiDoT Score* to explicitly measure the new challenge of bidirectional domain transferability. For each task $i$, the BiDoT score is defined as:

$$\text{BiDoT}^i = \frac{1}{|\mathcal{E}^{1:i}_{\text{unseen}}|} \sum_{(x,y) \in \mathcal{E}^{1:i}_{\text{unseen}}} \mathbf{1}(\hat{y} = y), \tag{10}$$

where the evaluation set $\mathcal{E}^{1:i}_{\text{unseen}}$ includes all previously learned classes tested on unseen domains $\mathcal{D}^{\text{unseen}}$:

$$\mathcal{E}^{1:i}_{\text{unseen}} = \bigcup_{k=1}^{i} \bigcup_{D^j \in \mathcal{D}^{\text{unseen}}} (C^k, D^j), \tag{11}$$

where class $C^k$ had no supervision during training. We report average (*A-BiDoT*) and final (*F-BiDoT*) scores. *A-BiDoT* captures consistent generalization, while *F-BiDoT* reflects overall generalization to unseen domains.

**Baselines.** Since our proposed setting, XIL, is novel and lacks directly comparable methods under identical conditions, we establish the most relevant prompt-based learning methods from CIL as baselines. These include traditional methods such as LwF (Li & Hoiem, 2017) and EWC (Kirkpatrick et al., 2017), and recent state-of-the-art prompt tuning methods: S-Prompts (Wang et al., 2022c), which learns independent prompts per domain; CODA-Prompt (Smith et al., 2023), which uses decomposed attention conditioned prompt tuning; and CPrompt (Gao et al., 2024), which trains the current task prompt and classifier guided by all existing prompts and classifiers. We also include SimpleCIL (Zhou et al., 2024a), a prototype-based classifier approach without further training of the pretrained model. To evaluate the inherent bidirectional domain transfer capability of the model, we additionally report results from offline joint fine-tuning (Joint-FT).

**Architecture.** All baselines were implemented using PyTorch (Paszke et al., 2019), with a ViT-B/16 backbone (Dosovitskiy et al., 2020) pretrained on ImageNet1K (Russakovsky et al., 2015). For high-quality synthetic image generation, we utilized IP-Adapter (Ye et al., 2023) built on top of Stable Diffusion XL (SDXL) (Podell et al., 2023).

**Training Details.** The prompt length $L$ was set to 5, for both XEED and S-Prompts across all datasets, and the denoising step $k$ was set to 50, yielding good image quality across all datasets. Other baseline hyperparameters followed official implementations (Li & Hoiem, 2017; Kirkpatrick et al., 2017; Wang et al., 2022c; Smith et al., 2023; Gao et al., 2024). To generate images, we extracted 5–10 class centroids from the training data and generated 25–30 samples per class.

## 5.2 EXPERIMENTAL RESULTS

We comprehensively analyzed our method, XEED, and baseline methods under the XIL setting from multiple perspectives. Table 1 reports performance on unseen domains and overall accuracy, demonstrating superior generalization of XEED and the limitations of existing methods. XEED achieved the highest BiDoT scores across all datasets, with up to a 31.41% improvement over the second-best baseline on PACS, and also showed superior accuracy on PACS and Office-31. The larger BiDoT gaps in datasets with high cross-domain variation, such as PACS and DomainNet, suggest that XEED effectively adapts semantics to evolving domains. Its strong standard accuracy further indicates effectiveness in both generalization and incremental learning. While CODA-P and

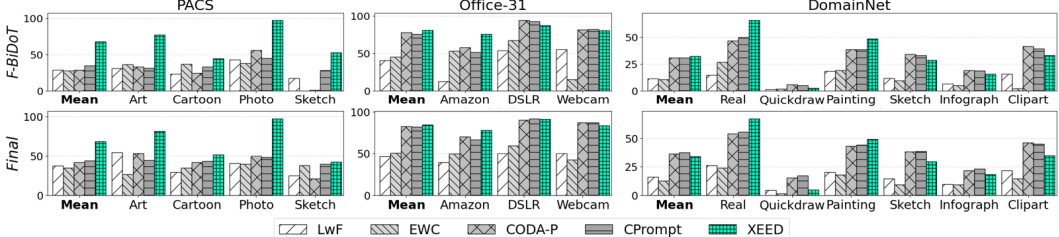

Figure 4: Per-domain accuracy gap between XIL and CIL in the final task. "Mean" indicates the average across domains. A more negative gap implies poorer cross-domain generalization.

Figure 5: Final BiDoT and accuracy per domain, averaged over three runs.

CPrompt focus on mitigating forgetting under static distributions, they showed limited generalization across all datasets. In Office-31, where in-domain variation is small, prototype-based methods, XEED and SimpleCIL, outperformed linear classifier methods, highlighting the advantage of prototype representations in improving generalization. These findings underscore the need for realistic settings, such as XIL, to drive progress beyond static continual learning scenarios.

To further assess the inherent ability of offline-trained Joint-FT and CIL baselines to transfer domain knowledge across tasks, we evaluated the final accuracy gap between XIL and CIL in Figures 2 and 4. As shown in Figure 2, baseline performance dropped significantly under XIL across all datasets, indicating limited generalization to unsupervised class-domain associations. In contrast, XEED achieved a positive gap with improved performance on PACS and showed smaller accuracy gaps overall. Figure 4 provides a detailed domain-wise comparison of accuracy gaps to examine whether methods exhibit domain-specific bias in generalization. For EWC, the accuracy gap in the Clipart domain of DomainNet was 52.6% below its mean, indicating strong domain-specific bias. In contrast, XEED showed only a 4.8% deviation in its worst case (Infograph), suggesting much more balanced generalization. Even in Office-31, CODA-P showed a 13.3% deviation in the Amazon domain, while XEED had only a 2.2% deviation in DSLR, indicating more stable performance. These results reveal limited bidirectional domain transferability and domain-wise bias in prior methods, underscoring the value of synthetic image generation and prototype-based classification for effective, unbiased domain adaptation.

Figure 5 shows final per-domain BiDoT scores and accuracies, highlighting generalization behavior under different method designs. Since domains follow task order, methods such as EWC, which use parameter regularization, showed degraded performance on later tasks across all datasets, as seen in their F-BiDoT scores. CODA-P, due to its decomposed prompt weighting mechanism, tended to overfit to specific domains, leading to imbalanced generalization across domains. CPrompt ad-

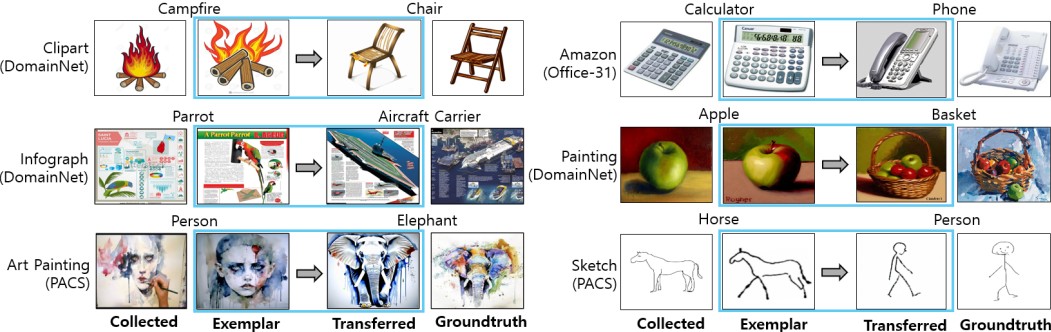

Figure 6: Visualization of generated samples. Exemplars were generated from collected data and used to synthesize cross-domain transferred images, shown alongside the ground truth for each class in the target domain.

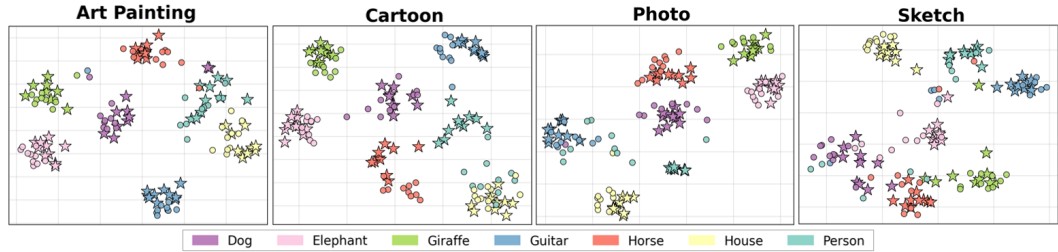

Figure 7: t-SNE visualization of CLIP embeddings on PACS. Stars indicate ground truth, circles represent generated data, and colors denote different classes.

dresses a key CIL limitation by learning prompts and classifiers beyond current-task data, outperforming CODA-P under standard CIL (Gao et al., 2024). However, it showed reduced performance on Office-31 in both F-BiDoT and final accuracy under XIL, suggesting that methods overly tailored to fixed CIL assumptions may struggle to generalize in more dynamic scenarios such as XIL. XEED leverages domain-specialized prompts and prototype-based classification to generalize without overfitting, consistently achieving the highest mean F-BiDoT across all domains and datasets.

To highlight the effectiveness of our generated samples in approximating real-world data, Figure 6 compares the cross-domain transferred data with the ground truth, while Figure 7 illustrates their proximity in feature space.

## 5.3 ABLATION STUDY

We conducted an ablation study to evaluate the contribution of each component in XEED, as shown in Table 2. Four ablation configurations were tested. First, **w/o prompt** removes the domain-specialized prompts, using a frozen pre-trained feature extractor without any domain adaptation. This setting evaluates whether prototype embeddings alone are sufficient to distinguish between classes across domains. Second, **w/o generation** excludes synthetic image generation, assessing the role of generated samples in maintaining robust bidirectional domain transferability. Third, **w/o prototype** replaces prototype-based classification with an auxiliary linear classifier, testing whether generalization is still achievable without semantic expansion using evolving prototypes. Lastly, **w/o inference** disables domain prototype matching and instead randomly selects a domain prompt during inference, measuring the importance of domain-aware prompt selection for consistent prediction.

*W/o prompts* led to moderate drops in F-BiDoT scores across all datasets, despite relatively stable final accuracy. This indicates that domain-specific prompts are crucial for achieving robust generalization to unseen domains, even when the base model performs reasonably well. *W/o generation* caused the most severe performance degradation, especially in F-BiDoT, demonstrating the central role of generated samples in enabling bidirectional domain transfer. *W/o prototype* led to a disproportionately larger degradation in F-BiDoT compared to final accuracy, more than in other ablations. This highlights the importance of evolving prototypes in enhancing generalization to unseen domains. Lastly, *w/o inference* showed minimal impact in Office-31 due to its low inter-domain variation, but led to larger drops in both F-BiDoT and final accuracy in more diverse datasets. This highlights the importance of domain-aware prompt selection under greater domain shift. Overall, the ablations confirm the necessity of each XEED component for robust domain transfer and incremental learning.

Table 2: Ablation study on XEED components.

| Method | PACS | | DomainNet | | Office-31 | |
|---|---|---|---|---|---|---|
| | F-BiDoT | Final | F-BiDoT | Final | F-BiDoT | Final |
| **XEED** | **65.19** | **61.86** | **33.63** | **35.30** | **78.08** | **80.72** |
| w/o prompts | 45.22 | 58.46 | 26.62 | 30.66 | 65.41 | 73.80 |
| w/o generation | 20.91 | 39.78 | 4.47 | 10.52 | 33.62 | 52.40 |
| w/o prototype | 18.85 | 43.48 | 5.24 | 15.05 | 35.60 | 54.81 |
| w/o inference | 52.33 | 51.26 | 26.00 | 26.54 | 77.83 | 79.78 |

## 5.4 ADDITIONAL EXPERIMENTAL ANALYSES

**Impact of Synthetic Exemplar Quality on Prototype Dynamics.** To assess whether synthetic exemplars introduce stylistically inconsistent samples that could distort prototype learning, we performed an additional analysis on the Office-31 dataset. For each class-domain pair, we used CLIP embeddings to compute Mahalanobis distances over generated samples, treating the bottom 5% as normal samples and the top 5% as potential outliers.

To assess the impact of such outliers on prototype formation, we compare prototype-normal distances (Normal) with prototype-outlier distances (Outlier). If outliers distorted the prototypes, they would lie closer to the prototype than normal samples, effectively pulling the prototype toward them. However, we consistently observe the opposite: as shown in Table 3,

Table 3: Prototype-sample distance statistics.

| Domain | Normal | Outlier | % Farther | GT Outlier |
|--------|--------|---------|-----------|------------|
| Amazon | 25.23 | 37.75 | 85.0 | 41.08 |
| DSLR | 19.88 | 29.89 | 100.0 | 23.53 |
| Webcam | 23.71 | 37.41 | 95.2 | 27.25 |

domain-level results averaged across all classes indicate that outliers remain farther from the prototype (% Farther), with 85-100% of classes exhibiting this pattern. Because prototype updates rely on mean aggregation, higher-quality samples naturally dominate the estimate, preventing atypical outliers from exerting significant influence.

Finally, we compare these distances with those from the real dataset. The absolute prototype-sample distances of synthetic images are comparable to the natural intra-class variation in real data (GT Outlier), and in some domains (e.g., Amazon), real outliers are even farther from the prototype. This shows that the variability introduced by generated samples falls within the natural range of real images and does not negatively affect prototype construction.

**Effect of Trainable Parameter on Domain Specialization.** XIL requires models to generalize to unseen class-domain combinations, and excessive trainable parameters can lead the model to over-specialize to domain-specific patterns present in the training distribution. For this reason, XEED employs a frozen backbone with lightweight domain prompts, which limits overfitting while preserving generalization capacity, and still provides strong performance on seen domains. To validate this design choice, we implemented an adapter-based variant by attaching residual adapters (Houlsby et al., 2019) to every ViT block and compared it with the proposed prompt-based XEED model on the Office-31 dataset. As shown in Table 4, the adapter variant shows an almost 11% drop in F-BiDoT, indicating weakened generalization to unseen class-domain combinations when more trainable parameters are introduced.

In contrast, XEED maintains higher Final and Avg performance, showing that lightweight prompt adaptation provides a better balance between robustness to distribution shift and strong in-domain accuracy.

Table 4: Trainable parameter comparison.

| Method | F-BiDoT | A-BiDoT | Final | Avg |
|--------|---------|---------|-------|-----|
| **Prompt (Ours)** | **78.08** | **73.67** | **80.72** | **83.19** |
| Adapter | 67.97 | 60.85 | 76.71 | 81.55 |

**More Incremental Tasks.** We further evaluate a more fine-grained setting by splitting each incremental task in half (6 tasks total, with 2 per domain in Office-31). As shown in Table 5, increased granularity substantially degrades the strongest baseline, highlighting that limited per-task class diversity leads to poorer generalization. XEED maintains strong generalization and remains superior overall.

Table 5: More Incremental Tasks.

| Method | F-BiDoT | A-BiDoT | Final | Avg |
|--------|---------|---------|-------|-----|
| **XEED** | **84.40** | **80.92** | **80.75** | **84.19** |
| CPrompt | 70.12 | 71.72 | 73.75 | 79.72 |

**Prompt Selection.** In XIL, classes and domains continually expand, and a parametric selector trained only on past distributions would struggle to generalize to unseen class-domain combinations. XEED uses a non-parametric selector in the prototype space, which includes synthetic exemplars for unseen combinations and enables flexible routing without task-specific parameters. Prompt selection accuracy on Office-31 (Table 6), measured on the final task, shows that the selector generalizes well even to unseen class-domain pairs. Note that

Table 6: Prompt selection.

| Metric | Accuracy (%) |
|--------|--------------|
| Overall | 86.86 |
| Unseen | 83.19 |

the goal is not domain classification, but effective routing to prompts that support class prediction.

## 6 CONCLUSION

In this work, we introduced Cross-Expanding Incremental Learning (XIL), a novel setting that extends class-incremental learning to enable bidirectional domain knowledge transfer. To tackle the challenges of XIL, we proposed XEED (Semantic Expansion through Evolving Domains), a framework that combines domain-specialized prompts, residual-guided modulation, and evolving prototypes. We introduced the BiDoT Score, a new metric for explicitly measuring bidirectional domain transferability, which is not captured by existing CIL benchmarks. Experiments on diverse benchmarks demonstrated the effectiveness of XEED and revealed the limitations of prior methods under XIL. We hope this work fosters further research toward generalizable continual learning paradigms.

## 7 ACKNOWLEDGEMENT

This work was supported by the Institute of Information & Communications Technology Planning & Evaluation (IITP) grant funded by the Korea government (MSIT) [RS-2021-II211341, Artificial Intelligence Graduate School Program (Chung-Ang University)].

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

# A   APPENDIX

We provide additional clarifications here in response to the key concerns raised during rebuttal. Other revisions have been incorporated into the main paper.

**Relationship Between XIL and Cross-Domain Continual Learning.**   Existing definitions of Cross-Domain Continual Learning (CDCL) differ across the literature. As summarized in Section 2.2, (Simon et al., 2022) defines CDCL as CIL with domain generalization, where each task provides multi-domain data and model learns domain-invariant features. In contrast, (De Carvalho et al., 2024; Weng et al., 2024) frame CDCL as class/task-incremental learning with domain adaptation, where models have access to unlabeled target domain data and adapt to a fixed target domain.

XIL differs from both: neither classes nor domains repeat across tasks, and each class appears in exactly one domain. Model receives neither multi-domain supervision nor target domain data, yet must predict all classes across all domains at test time. This requires domain knowledge to transfer across classes and to support both forward and backward transfer, capabilities not addressed by existing CDCL settings. This structure reflects practical scenarios where classes arise in new domains without additional supervision. XIL captures this challenge, and XEED is designed to enable the necessary bidirectional cross-class transfer of domain knowledge.

**Choice of Datasets for Evaluating XIL.**   XIL requires each task to introduce new class-domain combinations, which many common benchmarks cannot support. CoRE50 (Lomonaco & Maltoni, 2017) includes only 10 classes across 11 domains, allowing at most one class per task and preventing valid multi-class XIL task construction. Using subsets would make results highly dependent on the chosen sessions.

Likewise, benchmarks such as CIFAR-100 (Krizhevsky et al., 2009), TinyImageNet (Le & Yang, 2015), and ImageNet-subsets contain only a single domain, collapsing XIL into ordinary CIL. Multi-domain datasets like ImageNet-R (Hendrycks et al., 2021) are also unsuitable because domains are not aligned across classes, making unseen class-domain evaluation impossible.

For these reasons, we use Office-31, PACS, and DomainNet, which naturally satisfy XIL's structural requirements and provide meaningful evaluation of cross-class domain transfer.

**Effect of Domain Order.**   We evaluated XEED and CPrompt under randomly shuffled domain orders across both Office-31 and PACS, as shown in the table below. XEED remains stable across orderings, while CPrompt shows strong sensitivity. *Domain abbreviations: W - Webcam, A - Amazon, D - DSLR, C - Cartoon, S - Sketch, P - Photo, Ar - Art.*

| Dataset | Domain Order | Method | *F-BiDoT* | *A-BiDoT* | *Final* | *Avg* |
|---------|--------------|--------|-----------|-----------|---------|-------|
| Office-31 | W → A → D | **XEED** | **75.75** | **75.54** | **80.02** | **85.80** |
|  |  | CPrompt | 66.97 | 65.11 | 75.37 | 83.98 |
| Office-31 | D → W → A | **XEED** | **79.93** | **85.00** | **83.16** | **91.89** |
|  |  | CPrompt | 67.62 | 82.16 | 76.16 | 91.22 |
| PACS | C → S → P → Ar | **XEED** | **64.99** | **70.24** | **67.67** | **80.57** |
|  |  | CPrompt | 18.18 | 31.78 | 28.27 | 61.57 |
| PACS | S → Ar → P → C | **XEED** | **59.88** | **71.07** | **66.14** | **81.58** |
|  |  | CPrompt | 39.73 | 54.41 | 53.21 | 75.09 |

**Future Work: Toward a Unified Discriminative-Generative Model.**   Discriminative objectives (e.g., cross-entropy) focus on class-specific cues but often miss global distributional structure, limiting domain separation and transfer. Generative models such as diffusion models capture full data distributions and thus encode domain-level characteristics that discriminative training alone cannot. Our findings suggest that both signals are necessary for stable learning under domain shift.

No existing method unifies discriminative modeling with diffusion-based distribution modeling. As a future direction, we aim to develop a hybrid discriminative-generative framework building on XEED, enabling more efficient and robust forward and backward transfer of domain knowledge.

**XIL Task Construction and Evaluation Code.** We provide reference implementation code for (1) constructing XIL tasks and (2) computing the XIL evaluation metrics used in the main experiments. These snippets reproduce the core logic used in our experiments and are included for clarity and reproducibility. The full implementation is available at `https://github.com/heayounchoi/XIL`.

Code 1: Python code for XIL task construction.

```python
def construct_xil_tasks(class_list, domain_list):
    # Shuffle global class order
    classes = class_list.copy()
    random.shuffle(classes)

    # Classes allocated per domain
    per_domain = len(classes) // len(domain_list)

    task_order = []
    idx = 0

    # Assign class subsets to domains in order
    for _ in domain_list:
        task_order.append(classes[idx : idx + per_domain])
        idx += per_domain

    return task_order
```

Code 2: Python code for computing XIL metrics (Final, BiDoT, and Prompt Accuracy).

```python
def compute_xil_metrics(y_pred, y_true, dom_pred, dom_true, task_order):
    metrics = {}

    # 1. Final accuracy
    metrics["Final"] = 100 * (y_pred == y_true).mean()

    # 2. BiDoT (unseen class-domain accuracy)
    unseen_idx = [
        i for i in range(len(y_true))
        if y_true[i] not in task_order[int(dom_true[i])]
    ]
    metrics["BiDoT"] = 100 * (y_pred[unseen_idx] == y_true[unseen_idx]).mean()

    # 3. Prompt selection accuracy
    metrics["Prompt"] = 100 * (dom_pred == dom_true).mean()

    return metrics
```

