# OpenReview forum: "XIL: Cross-Expanding Incremental Learning"
_ICLR.cc/2026/Conference — ICLR 2026 Poster_

### Official Review · Reviewer_EJAk · 2025-10-19

**Soundness:** 2
**Presentation:** 3
**Contribution:** 2
**Rating:** 4
**Confidence:** 5

**Summary:**

The paper introduces Cross-Expanding Incremental Learning (XIL)—a setting where both classes and domains evolve—and proposes XEED, a rehearsal-free framework that (a) learns domain-specialized prompts, (b) modulates class features with domain residuals using a diffusion model (IP-Adapter + SDXL), and (c) performs prototype-based classification with evolving, domain-aware prototypes. Also, define the BiDoT Score to measure bidirectional domain transferability. XEED substantially improves BiDoT and accuracy on PACS, Office-31, and DomainNet versus strong CIL/prompt baselines.

**Strengths:**

The paper introduces a new continual learning setting, specifically,

1) Clean formulation of XIL + dedicated BiDoT metric.

2) Method components are modular and leverage frozen encoder + small learned parts.

3) Privacy-friendlier than exemplar replay; aligns with synthetic replay trends.

**Weaknesses:**

1) The paper admits that no directly comparable XIL baselines exist and hence reuses CIL prompt-based methods (S-Prompts, CODA-P, CPrompt, etc.) as proxies. Since XIL is a new setting, the paper adapts existing CIL methods as baselines rather than comparing with purpose-built domain-evolving or generative continual learning frameworks. This makes it difficult to fully assess XEED’s relative performance or to verify whether its improvements come from the new setting or the framework design itself.

2) XEED’s strongest BiDoT improvements depend critically on synthetic image generation using IP-Adapter + SDXL. The ablation study shows that removing this component causes F-BiDoT to plummet (e.g., from 65.19 → 20.91 on PACS and 33.63 → 4.47 on DomainNet). This dependence implies that the system’s success hinges on access to a large diffusion model and high-quality generation. The method may fail or become impractical if the computing is limited or the diffusion model struggles with domain realism.

3) Limited or no experiment in common continual learning benchmarks such as CIFAR100, TinyImagenet200, ImageNetsubset100, Food, Cars, ImageNet-R/A.

**Questions:**

Please refer to the weakness

---

> ### Author Response · Authors · 2025-11-20
> **Official Response to Reviewer EJAk - Comment #1**
>
> We thank the reviewer for the thoughtful feedback and for highlighting the **clear formulation of XIL**, the **contribution of the BiDoT metric**, the **modularity of the proposed method**, and its **privacy-friendly** use of synthetic replay. We appreciate these positive remarks and address the remaining concerns below. All clarifications and supplementary analyses from our response are included in the Discussion section of the paper’s Appendix.
>
> 1. Justification of Baseline Selection Under the New XIL Setting
>     - We appreciate the reviewer’s concern and clarify that Sec. 2.2 (More Problem Settings in CIL) of the paper systematically reviews all **purpose-built domain-evolving continual learning settings** and explains why none of them can be directly applied to XIL. To reiterate the key distinctions:
>         - **Class-incremental domain adaptation** methods require **labeled target-domain data for both old and new classes**.
>         - **Cross-Domain Continual Learning (CDCL)** methods learn **domain-invariant features from multi-source domain supervision**.
>         - Furthermore, the **CDCL setting referenced by Reviewer w46G** [R1] assumes that the **model always has access to unlabeled target-domain samples during training**.
>     - All of these approaches are built on assumptions that rely on multi-domain supervision or access to target-domain data for adaptation or domain generalization. In XIL, however, neither classes nor domains recur across tasks, and **each class appears in exactly one source domain**. Consequently, when these methods are applied directly to XIL, they **cannot function as intended**: the necessary data sources simply do not exist under the XIL setting, and thus these methods cannot serve as meaningful baselines.
>     - In addition, one might argue that, because XEED employs a generative model, generative replay would be the most appropriate baseline family. However, as shown in the table below, **recent literature consistently shows that prompt-based CIL methods outperform generative replay approaches on the same benchmark datasets**. Consequently, we adopt **state-of-the-art prompt-based methods** as baselines rather than generative replay approaches, ensuring that XEED is compared against the strongest available CIL techniques. To make this clear, we provide a summary of published results on the most commonly used CIL benchmark, CIFAR-100:
>
>
>         | Method Type | Performance on 10-task CIFAR-100 |
>         | --- | --- |
>         | **Replay-based** [R2] | 58.40% |
>         | **Prompt-based** [R3] | 87.82% |
>     - Importantly, **the improvements achieved by XEED do not stem from the new setting alone.** Existing CIL approaches fundamentally do not model or transfer domain knowledge **across classes**. When this assumption is violated in XIL—where substantial domain variation occurs and classes must generalize to unseen class–domain combinations—their performance decreases sharply, even when used exactly as originally designed (see Fig. 2). **Our contribution is to reveal this distributional fragility** and to show that XEED remains robust on unseen class–domain pairs where state-of-the-art CIL baselines fail, demonstrating that **XEED’s performance gains stem from its design rather than from the new setting itself.**

---

> ### Author Response · Authors · 2025-11-20
> **Official Response to Reviewer EJAk - Comment #2 & #3**
>
> 2. Practicality Under Limited Compute and Challenging Domains
>     - We appreciate the reviewer’s concern regarding the use of diffusion models. To ensure practicality under limited computing resources, **our design deliberately keeps the generation cost modest**. As detailed in Sec. 5.1, we use only 50 denoising steps, which requires roughly 12 seconds per class on a single A5000 GPU, and each class needs only 25–30 synthetic exemplars. Despite this lightweight setup, XEED achieves **up to a 31.41% improvement in F-BiDoT compared to the second-best baseline** (Table 1-PACS), indicating that the trade-off is both reasonable and effective—and that **the generated samples provide sufficient domain realism** to support robust domain transfer.
>     - Our empirical analysis indicates that relying solely on discriminative objectives (e.g., cross-entropy)—the dominant practice in recent continual learning research—**is insufficient for robustly disentangling and transferring domain knowledge**. Discriminative models tend to focus on fine-grained, class-specific cues that refine decision boundaries, yet they often fail to capture broader distributional properties. In contrast, generative models such as diffusion models optimize for reconstructing or approximating the full data distribution, allowing them to learn global, domain-level characteristics that purely discriminative training typically overlooks. For stable discriminative learning and reliable domain separation, we argue that **both forms of information are necessary**. To the best of our knowledge, **no existing work explicitly unifies discriminative modeling with diffusion-based distribution learning within a unified architecture**, making this a **promising direction for future research**. Building on this gap, we plan to extend XEED’s principles toward a **lightweight discriminative–generative hybrid model** that supports bidirectional domain transfer with substantially lower computational cost, ultimately replacing diffusion models without sacrificing performance.
>
> 3. On the Choice of Benchmarks for Evaluating XIL
>     - We appreciate the reviewer’s suggestion to include results on standard continual learning benchmarks such as CIFAR-100, TinyImageNet, and ImageNet-subsets. **DomainNet is also widely used in incremental learning [R3, R4], and we already report results on this dataset in the paper.** However, the datasets mentioned by the reviewer are fundamentally incompatible with the goals of the XIL setting.
>     - Benchmarks like CIFAR-100, TinyImageNet, and ImageNet-subsets contain **only a single domain**, meaning that no domain shift occurs across tasks. XIL, by design, assumes that each incremental step introduces a new domain, enabling evaluation of cross-class domain transfer. Using single-domain datasets would collapse XIL into ordinary CIL—the very assumption we argue is insufficient—thereby removing the core challenge the setting is intended to capture. Consequently, these benchmarks cannot meaningfully evaluate XIL.
>     - We also examined multi-domain datasets such as ImageNet-R. Although ImageNet-R includes multiple style domains, these domains are **not aligned across classes**: each class appears in only a subset of domains, and many class–domain pairs are missing entirely. As a result, evaluating generalization to unseen class–domain combinations—an essential requirement of XIL—is impossible.
>     - Furthermore, XEED is architecturally designed for **cross-class domain transfer**, rather than maximizing accuracy on fixed, single-domain distributions. Its inductive biases improve robustness under domain-evolving conditions but are not necessarily aligned with conventional CIL benchmarks, which are purpose-built for single-domain scenarios.
>     - For these reasons, the datasets used in our work—Office-31, PACS, and DomainNet—are the appropriate and necessary choices. They naturally satisfy the assumptions of XIL and accurately reflect real-world scenarios in which both classes and domains evolve over time. We will add a clarification on the dataset requirements for evaluating XIL to Sec. 5.1 (Experimental Setup) in the revised manuscript.
>
> References:
>
> [R1] Towards Cross-Domain Continual Learning (ICDE 2024)
>
> [R2] DiffClass: Diffusion-Based Class Incremental Learning (ECCV 2024)
>
> [R3] Consistent Prompting for Rehearsal-Free Continual Learning (CVPR 2024)
>
> [R4] CODA-Prompt: COntinual Decomposed Attention-based Prompting for Rehearsal-Free Continual Learning (CVPR 2023)

---

> ### Author Response · Authors · 2025-11-25
> **A Note of Thanks**
>
> Thank you for the time and effort you put into reviewing our work. Your thoughtful comments and questions have been very helpful in improving the manuscript and clarifying its contribution. If anything else comes to mind, please feel free to share it. Wishing you a relaxing end of the year and a joyful start to the new one!

---

> ### Author Response · Authors · 2025-12-02
> **Revision History of Official Review of Submission10066 by Reviewer EJAk**
>
> Instead of leaving a separate comment that their concerns were resolved, the reviewer updated their review on 25 Nov 2025 as follows:
>
> -------
> Questions:
> Please refer to the weakness.
>
> UPDATES:
>
> Thanks for addressing the comments. It is clearer now. I am raising my score. Good luck :).

---

### Official Review · Reviewer_w46G · 2025-10-28

**Soundness:** 2
**Presentation:** 3
**Contribution:** 2
**Rating:** 4
**Confidence:** 4

**Summary:**

This paper presents cross-expanding incremental learning where not only class changes occur but also domain shift happens at the same time. To address this problem, authors proposes Semantic Expansion through Evolving Domains (XEED).

**Strengths:**

1) although it intersects with existing setting, the XIL problem is new.
2) the proposed solution, namely XEED, performs well.

**Weaknesses:**

1) the following references also deal with domain adaptation and continual learning. I suggest to review them in the paper.

[1] Towards Cross-Domain Continual Learning
[2] Cross-Domain Continual Learning via CLAMP

2) please kindly explain the difference of your works with cross-domain continual learning as proposed in [1] and [2]

3) I suggest to elaborate more on the real-world context of your setting. This allows readers to appreciate more on your contributions. Also, it is suggested to link directly your problem with a concrete dataset that you use. Is there any dataset that can represent your problem?

4) the prompt selection mechanism is non-parametric and perhaps over-simplified.

5) prompt selection accuracy should be reported.

6) the domain order should affect the result. it should be detailed in the paper.

**Questions:**

1) the following references also deal with domain adaptation and continual learning. I suggest to review them in the paper.

[1] Towards Cross-Domain Continual Learning
[2] Cross-Domain Continual Learning via CLAMP

2) please kindly explain the difference of your works with cross-domain continual learning as proposed in [1] and [2]

3) I suggest to elaborate more on the real-world context of your setting. This allows readers to appreciate more on your contributions. Also, it is suggested to link directly your problem with a concrete dataset that you use. Is there any dataset that can represent your problem?

4) the prompt selection mechanism is non-parametric and perhaps over-simplified.

5) prompt selection accuracy should be reported.

6) the domain order should affect the result. it should be detailed in the paper.

---

> ### Author Response · Authors · 2025-11-20
> **Official Response to Reviewer w46G - Comment #1**
>
> We thank the reviewer for the thoughtful feedback and for recognizing that, XIL introduces **a new problem formulation**, and that our proposed solution, XEED, demonstrates **strong performance** under this challenging setup. We appreciate these encouraging remarks and address all remaining concerns in the response below. All clarifications and supplementary analyses from our response are included in the Discussion section of the paper’s Appendix.
>
> 1. How XIL Differs from Cross-Domain Continual Learning
>     - We thank the reviewer for pointing us to Towards Cross-Domain Continual Learning and Cross-Domain Continual Learning via CLAMP. While these works also address domain-related continual learning, their problem formulation is fundamentally different from the XIL setting proposed in our paper. We will add a discussion of these works to the Introduction of the revised manuscript, where we review problem settings that address domain-gap mismatch in incremental scenarios (Lines 63-74).
>     - In Sec. 2.2 (More Problem Settings in CIL), our work already discusses Cross-Domain Continual Learning (CDCL) as defined in [R1]. That setting corresponds to class-incremental learning combined with domain generalization: classes do not overlap across tasks, **each task provides multi-domain data for those classes**, and at test time the model is asked to generalize to a domain that was never seen during training. Because source tasks include multiple domains, models can explicitly learn domain-invariant features, and the problem is positioned within the domain generalization paradigm.
>     - In contrast, the works referenced by the reviewer, defines CDCL as class/task-incremental learning combined with domain adaptation. In this setting, classes across tasks may or may not overlap, and the **model always has access to unlabeled target-domain samples during training**. The learning objective is adaptation to a fixed target domain, and thus these methods operate within the domain adaptation family, where domain shifts can be mitigated through direct exposure to the target-domain distribution.
>     - The proposed XIL setting differs substantially from both. In XIL, neither classes nor domains repeat across tasks, and **each class is observed in exactly one source domain**. There is no opportunity to learn domain-invariant features from multi-domain supervision, nor is any target-domain data provided during training. At test time, the model must make predictions for **all** previously seen domains on **all** classes, despite never observing multi-domain data for any class. This structure forces the model to transfer domain knowledge **across classes**, a capability that prior CDCL methods are not designed to support. Moreover, those settings assume predominantly **forward-only** knowledge transfer, whereas XIL requires both **forward and backward** transfer of domain information across evolving class–domain combinations.
>     **We believe this makes XIL a more realistic formulation**: in practical deployment, systems often face new classes appearing in entirely new domains, without access to multi-domain supervision or target-domain samples during training. In such scenarios, models must **reuse and transfer** domain knowledge acquired from previous tasks—rather than rely on domain-invariant learning or direct domain adaptation—which is precisely the challenge defined by XIL. Consequently, XIL represents a new and more demanding setting that existing CDCL methods cannot handle, and XEED is specifically designed to support the required **bidirectional domain knowledge transfer**.

---

> ### Author Response · Authors · 2025-11-20
> **Official Response to Reviewer w46G - Comment #2 & #3**
>
> 2. Practical Relevance of XIL and Dataset Justification
>     - We appreciate the reviewer’s suggestion to further elaborate on the real-world motivation behind XIL and to clarify how it relates to the datasets used in our experiments. **The real-world scenario introduced in the paper (lines 73-81) is directly reflected in the domain characteristics present in our benchmarks.** Datasets such as Office-31 contain high-quality controlled images (e.g., Amazon) as well as operator-provided or in-the-wild captures (e.g., DSLR, Webcam). PACS and DomainNet further extend this diversity by including hand-drawn sketches (e.g., Sketch, Quickdraw), technical diagrams (e.g., Infograph), and naturally collected photographs (e.g., Photo, Real), thereby covering a wide spectrum of domain variations ranging from stylized illustrations to uncontrolled real-world imagery. These datasets collectively represent the types of domain shifts that arise organically in practice and align closely with the goals of the XIL setting.
>     - An additional real-world scenario that closely reflects XIL is the **continuous data collection pipeline of services like Google Maps**. Vehicles and drones capture images under constantly changing conditions—day/night cycles, seasons, weather, and camera variations—while the set of observable objects also changes across regions. Rural areas may suddenly include newly developed urban structures, and urban outskirts may contain agricultural machinery or natural terrain. Because class and domain shift together in unpredictable ways, models cannot rely on seeing multiple domains per class or accessing target-domain samples during training; they must instead reuse and transfer domain knowledge acquired from prior tasks. This is exactly the challenge that XIL is designed to capture.
>     - Because no existing benchmark fully captures this dynamic and reversible interplay between class and domain, we construct XIL using multi-domain datasets such as Office-31, PACS, and DomainNet. These datasets naturally encode the diverse domain variations, making them a suitable and realistic foundation for evaluating methods under the XIL setting.
>
> 3. On the Design and Evaluation of the Prompt Selection Mechanism
>     - We appreciate the reviewer’s concern that our prompt selection mechanism may appear overly simple due to its non-parametric design. However, this choice is intentional: in XIL, both classes and domains continually expand, and **a parametric selector would necessarily be trained only on the distributions observed so far, limiting its ability to generalize to unseen class–domain combinations**. In contrast, our non-parametric mechanism operates directly in a prototype space that includes synthetic exemplars for unseen class–domain combinations, enabling flexible routing without introducing task-specific parameters. This design allows the selector to naturally generalize to novel combinations that were never observed during training.
>     - To address the reviewer’s suggestion, we additionally report the prompt selection accuracy, which quantifies how often our selection mechanism correctly identifies the ground-truth domain for each input. We compute this accuracy on the final task, after the entire sequence has been learned. The results on the Office-31 dataset below show that our non-parametric routing mechanism achieves high accuracy despite its simplicity. Notably, **it maintains strong performance even on class–domain combinations that were unseen during training,** indicating that the selector can generalize to novel combinations beyond the training distribution. This analysis will be included in Sec. 5.4 (Additional Experimental Analyses) of the revised manuscript.
>
>
>         | Metric | Prompt Accuracy (%) |
>         | --- | --- |
>         | Overall | 86.86 |
>         | Unseen | 83.19 |
>     - Importantly, the purpose of prompt selection is not to perform domain classification, but to route each input to the prompt that best facilitates class prediction under unseen class–domain combinations. The substantial gains in F-BiDoT and A-BiDoT reported in the paper confirm that the selected prompts effectively support class prediction in practice.

---

> ### Author Response · Authors · 2025-11-20
> **Official Response to Reviewer w46G - Comment #4**
>
> 4. Effect of Domain Order
>     - We thank the reviewer for raising the concern regarding the potential sensitivity of our method to domain order. To examine this, we conducted additional experiments on Office-31 and PACS, evaluating XEED and the strongest baseline, CPrompt, under **multiple randomly shuffled domain sequences**. Across both datasets, **XEED remains consistently robust under different domain orders**, whereas the baseline exhibits substantial sensitivity to the ordering. The results are grouped by dataset and domain order below.
>
>         ***
>         **Office-31**
>
>         *Domain Order: webcam → amazon → dslr*
>
>         | Method | F-BiDoT | A-BiDoT | Final | Avg |
>         | --- | --- | --- | --- | --- |
>         | **XEED (Ours)** | **75.75** | **75.54** | **80.02** | **85.80** |
>         | CPrompt | 66.97 | 65.11 | 75.37 | 83.98 |
>
>         *Domain Order: dslr → webcam → amazon*
>
>         | Method | F-BiDoT | A-BiDoT | Final | Avg |
>         | --- | --- | --- | --- | --- |
>         | **XEED (Ours)** | **79.93** | **85.00** | **83.16** | **91.89** |
>         | CPrompt | 67.62 | 82.16 | 76.16 | 91.22 |
>         ***
>         **PACS**
>
>         *Domain Order: cartoon → sketch → photo → art*
>
>         | Method | F-BiDoT | A-BiDoT | Final | Avg |
>         | --- | --- | --- | --- | --- |
>         | **XEED (Ours)** | **64.99** | **70.24** | **67.67** | **80.57** |
>         | CPrompt | 18.18 | 31.78 | 28.27 | 61.57 |
>
>         *Domain Order: sketch → art → photo → cartoon*
>
>         | Method | F-BiDoT | A-BiDoT | Final | Avg |
>         | --- | --- | --- | --- | --- |
>         | **XEED (Ours)** | **59.88** | **71.07** | **66.14** | **81.58** |
>         | CPrompt | 39.73 | 54.41 | 53.21 | 75.09 |
>
> References:
>
> [R1] On Generalizing Beyond Domains in Cross-Domain Continual Learning (CVPR 2022)

---

> > ### Comment · Reviewer_w46G · 2025-11-21
> > **Thank you for your clarification**
> >
> > Thank you for your clarifications. This paper is now stronger in my view. I agree to increase my score.

---

> > > ### Author Response · Authors · 2025-11-21
> > > **A Note of Thanks**
> > >
> > > Thank you for the time and effort you put into reviewing our work.
> > > Your thoughtful requests for clarification and additional experiments helped us strengthen the paper and reflect more deeply on its broader implications.
> > > If anything else comes to mind, please feel free to leave another comment.
> > > Wishing you a wonderful end of the year and a happy new year!

---

### Official Review · Reviewer_gcQm · 2025-10-29

**Soundness:** 2
**Presentation:** 3
**Contribution:** 2
**Rating:** 4
**Confidence:** 3

**Summary:**

This paper proposes the Cross-Expanding Incremental Learning (XIL), a new setting of continual learning. The XIL presents the problem of existing setting of class-incremental learning (CIL), where the model learns from the data of same domains. XIL emphasizes the  capability of transferring knowledge across different domains and proposes a new settiing. A corresponding method XEED is proposed to address the XIL. XEED is a generative replay-based method which leverages a pre-trained diffusion model and CLIP to generate samples that are transferable across different domains. Domain-specific prompts and prototypes are proposed to preserve knowledge from different tasks. Several experiments are conducted to validate the effectness of XEED.

**Strengths:**

1. This paper consider a new paradigm of continual learning, where the capability of transferring across different domains are considered.
2. This paper is well writen with good presentation and clear motivation.
3. The technique of generating exemplars with representation modulation is noval and sound.

**Weaknesses:**

1. Using the pre-trained diffusion model and CLIP. Although this paper is overall good, I have several concerns about using the pre-trained diffusion model. Since a good exemplar or sample for transferring can be generated, the knowledge related to the domains and classes is already encoded in the pre-trained model. As such, do we still need continual learning of another model? Most existing generative replay-based methods train a generative model during the continual learning, which is incremental learning process parallel to the training model. However, if the model needs another model which already has the knowledge of future data, since we already have a good one, what is the meaning of incremental learning of that model?
2. The proposed method XEED is simple and preserving prompts and prototypes specific to distinct domains or classes has been widey studied in the community of continual learning.
3. The capability of transferring across domains, the major challenge defined in this paper, seems mainly benefit from the generative replay. It seems that the major problem defined in this paper is solved by introducing knowledge of other models while the other parts of this method have mere innovation. One can argue that with such a replay mechamism, existing CIL methods can be easily transferred to the setting of XIL.
4. Results on dataset specially designed for the domain and class incremental learning, e.g., CoRE50 [1]. should be included. The datasets used in the experiments are not designed for incremental learning. The CoRE50, which is designed for both the CIL and Domain-incremental learning tasks, can be a good benchmark of XIL.

[1] Vincenzo Lomonaco and Davide Maltoni. Core50: a new dataset and benchmark for continuous object recognition. In Conference on Robot Learning, pages 17–26, 2017.

**Questions:**

Please refer to the weaknesses.

---

> ### Author Response · Authors · 2025-11-20
> **Official Response to Reviewer gcQm - Comment #1**
>
> We thank the reviewer for the thoughtful feedback and for recognizing our work as introducing **“a new paradigm of continual learning,”** as well as noting that the paper is **well written with clear motivation** and that our representation-modulated exemplar generation is **“novel and sound.”** We appreciate these encouraging comments and will address all remaining concerns in the response below. All clarifications and supplementary analyses from our response are included in the Discussion section of the paper’s Appendix.
>
> 1. Clarifying the Role of the Pre-Trained Models
>     - Thank you for raising this thoughtful question regarding the use of pre-trained models. We would like to clarify that leveraging pre-trained models does not eliminate the need for continual learning; rather, it aligns with how recent continual learning research increasingly incorporates large-scale pre-trained models to enhance reasoning, representation, and inference, much like humans draw on accumulated perceptual and cognitive priors when learning new concepts.
>     - First, many recent continual learning frameworks already rely on pre-trained components. For example, most task models use pre-trained vision backbones for their strong representation capabilities. Likewise, existing generative-replay methods [R1] based on diffusion models also use pre-trained diffusion models, as their effectiveness stems from the **powerful generative capacity** acquired from large-scale data. In XEED, we similarly utilize this generative prior not to “know” future tasks, but to disentangle and extract **domain knowledge** that is otherwise unavailable to the task model.
>     - Similarly, the CLIP encoder is used purely as a general-purpose representation module, providing broad semantic priors rather than task-specific or future-task knowledge. For similar reasons, methods like [R2] employ a pretrained language model to generate text descriptions, not because it contains knowledge of future tasks, but because such pretrained models offer generic semantic structure that improves continual learning.
>     - In this sense, XEED does not bypass continual learning; rather, it leverages widely used pre-trained priors to disentangle domain information, while the task model still must incrementally acquire new semantic concepts and transfer domain knowledge across them.

---

> ### Author Response · Authors · 2025-11-20
> **Official Response to Reviewer gcQm - Comment #2**
>
> 2. Building Upon Widely Studied Strategies
>     - We appreciate the reviewer’s observation and agree that prompts and prototypes, when considered independently, are not new. However, the XIL setting introduces challenges that prior work has not addressed—**how prompts should be designed to transfer domain knowledge across classes, how prototypes should be constructed for unseen class–domain combinations, and how forward and backward domain transfer can be achieved simultaneously within a continual learning framework**. XEED is the first to operationalize these capabilities in a unified architecture, marking an architectural direction not explored in earlier literature.
>     - Recent prompt-based approaches typically increase the number of trainable parameters to capture task-specific variations. While effective for in-domain fitting, this trend results in **poor generalization to unseen domains**, as demonstrated in our experiments addressing **Reviewer RYCp’s question on “3. Adaptability Under Severe Domain Shifts.”** To examine how the amount of trainable parameters affects unseen-domain generalization, we implemented an **adapter-based variant of XEED** by inserting residual adapters after each block in the ViT backbone. Despite using significantly more trainable parameters, the adapter-based model fails to generalize to unseen class–domain combinations, exhibiting an **11% drop in F-BiDoT**. This demonstrates that simply adding parameter-efficient tuning mechanisms does not resolve the core challenge in XIL. The results on Office-31 are shown below:
>
>
>         | Method | F-BiDoT | A-BiDoT | Final | Avg |
>         | --- | --- | --- | --- | --- |
>         | **Original (Ours)** | **78.08** | **73.67** | **80.72** | **83.19** |
>         | **Adapter Variant** | 67.97 | 60.85 | 76.71 | 81.55 |
>     - Additionally, **how prototypes should be constructed for unseen class–domain combinations** is a problem that prior continual learning methods have not explored. In XEED, we address this by generating prototypes through **representation modulation**, enabling prototypes to generalize beyond observed domains.
>     - Moreover, existing CIL methods are designed solely for **forward knowledge transfer**, relying primarily on linear classifiers rather than prototype-based ones. As a result, they do not provide mechanisms for **bidirectional (forward + backward) domain knowledge transfer**, nor do they consider how newly acquired domain knowledge can propagate back to previously learned classes.
>     - For these reasons, XIL calls for **structural capabilities** that are absent in prior CIL approaches, and XEED is the first framework that systematically enables **bidirectional domain knowledge transfer across evolving class–domain combinations.**

---

> ### Author Response · Authors · 2025-11-20
> **Official Response to Reviewer gcQm - Comment #3 & #4**
>
> 3. Beyond Generative Replay in Addressing XIL
>     - We appreciate the reviewer’s insightful question regarding whether generative replay alone could allow existing CIL methods to transfer into the XIL setting. However, **generative replay by itself is insufficient for domain knowledge transfer**, because replayed images still entangle class-specific and domain-specific factors. In XEED, **representation modulation is essential** for explicitly separating domain attributes from class semantics, which is what enables bidirectional transfer across unseen class–domain combinations. Without this disentanglement step, replayed samples cannot support domain-level generalization.
>     - Although one might view this representation modulation as the dominant component, this does not mean that existing CIL methods can be applied to XIL simply by adding the component. As we discuss in **2. Building Upon Widely Studied Strategies**, XIL requires **structural capabilities** that existing CIL approaches do not provide—particularly in how prompts, prototypes, and domain information are structured for cross-class domain knowledge transfer. XEED is the first framework to systematically enable **bidirectional domain knowledge transfer across evolving class–domain combinations**, which cannot be achieved through the representation modulation component alone.
>     - Our empirical results further show that relying solely on discriminative objectives (e.g., cross-entropy)—a common practice in recent continual learning work—**does not adequately disentangle or transfer domain knowledge**. Discriminative models highlight class-specific, boundary-sharpening cues, yet they tend to ignore broader distributional structure. In contrast, generative approaches such as diffusion models learn to approximate the full data distribution, capturing global, domain-level characteristics that discriminative training typically overlooks. For stable discriminative performance and reliable domain separation, **both forms of information are necessary**. To the best of our knowledge, **no existing method explicitly integrates discriminative modeling with diffusion-based distribution learning**, highlighting an **important opportunity for further research**. Motivated by this gap, we aim to develop a **discriminative–generative hybrid model** grounded in XEED’s core principles to support efficient forward and backward transfer of domain knowledge.
>
>
> 4. On the Choice of Datasets for Evaluating XIL
>     - We appreciate the reviewer’s suggestion and agree that CoRE50 is an important benchmark for domain- and class-incremental learning. However, **CoRE50 is structurally incompatible with the XIL setting**, which requires each task to introduce new class–domain combinations. CoRE50 contains only 10 classes across 11 domain conditions; therefore, using the full dataset would allow **at most one class per task**, making XIL evaluation impossible. This is particularly problematic because state-of-the-art CIL baselines, such as those we compare against, require at least two classes per task to train a meaningful classifier.
>     - One could restrict CoRE50 to a subset of sessions, but **the results would depend heavily on which subset is chosen**, introducing undesirable evaluation bias. For this reason, we instead rely on datasets that naturally satisfy the structural requirements of XIL. DomainNet, widely adopted in incremental learning [R3, R4], provides rich class–domain diversity, and Office-31 offers **domain shifts highly similar to CoRE50** (e.g., background/lighting variation) while still enabling valid multi-class task construction. The strong results on Office-31 therefore **already demonstrate XEED’s robustness under CoRE50-like domain shifts**, but within a dataset design that properly supports XIL. We will add a clarification on the dataset requirements for evaluating XIL to Sec. 5.1 (Experimental Setup) in the revised manuscript.
>
> References:
>
> [R1] DiffClass: Diffusion-Based Class Incremental Learning (ECCV 2024)
>
> [R2] Convolutional Prompting meets Language Models for Continual Learning (CVPR 2024)
>
> [R3] Consistent Prompting for Rehearsal-Free Continual Learning (CVPR 2024)
>
> [R4] CODA-Prompt: COntinual Decomposed Attention-based Prompting for Rehearsal-Free Continual Learning (CVPR 2023)

---

> > ### Comment · Reviewer_gcQm · 2025-11-28
> >
> > Thank you for your response! Most of my concerns are well addressed and I wound like to increase the score to 6. However, the system does not allow me to edit previous review and score at present. Onced allowed, I will raise the score and I hope AC can refer to the updated review and score. Thank you!

---

> > > ### Author Response · Authors · 2025-11-28
> > > **A Note of Thanks**
> > >
> > > Thank you for the time and effort you put into reviewing our work. Your thoughtful comments and questions have been very helpful in improving the manuscript and clarifying its contribution. We are genuinely sorry that ICLR has closed the discussion phase, as we would have welcomed the opportunity to continue the exchange. If any additional comments come to mind, we would be very grateful if you could share them—either through the system, if further comments are still possible, or later on by email after the review process has ended. Wishing you a relaxing end of the year and a joyful start to the new one!

---

### Official Review · Reviewer_9JFF · 2025-10-30

**Soundness:** 3
**Presentation:** 3
**Contribution:** 3
**Rating:** 6
**Confidence:** 3

**Summary:**

This paper introduces Cross-Expanding Incremental Learning (XIL), a new, more realistic problem setting that extends traditional Class-Incremental Learning (CIL) by requiring models to learn new classes that arrive from distinct, evolving domains. The core challenge is achieving bidirectional domain transferability (BiDoT), where the model must generalize previously learned classes to new domains and new classes to old domains, even for combinations never seen during training. To address this, the authors propose a novel framework called Semantic Expansion through Evolving Domains (XEED), which utilizes domain-specialized prompts to adapt to different domains, a generative model to synthesize images for unseen class-domain pairs, and evolving prototypes to continuously expand the classifier's semantic space. Accompanied by a new evaluation metric, the BiDoT Score, experiments show that XEED significantly outperforms existing CIL methods, which struggle in this dynamic setting, thereby establishing a strong foundation for continual learning in real-world, non-static environments.

**Strengths:**

The paper's foremost strength is its introduction of the Cross-Expanding Incremental Learning (XIL) problem, which establishes a more realistic and challenging benchmark for continual learning by forcing models to simultaneously handle new classes and evolving data domains.

The proposed XEED presents a highly innovative solution, ingeniously using a generative model to synthesize data for unseen class-domain pairs, thereby directly addressing the core challenge of bidirectional knowledge transfer.

**Weaknesses:**

The paper does not sufficiently explain how the baseline methods, originally designed for the standard CIL setting, were adapted to the new multi-domain XIL setting. This is particularly concerning for methods like S-Prompts, which show drastically lower performance, raising questions about whether the comparison is entirely fair

As a paper proposing a new problem setting and a novel evaluation metric, the absence of publicly available code is a major drawback. This lack of resources makes it extremely difficult for the research community to verify the reported results, adopt the XIL benchmark, and fairly compare future methods, thus hindering follow-up research.

While the XEED is effective, its core components: domain-specific prompts, generative replay, and prototype-based classifiers, are existing techniques.

**Questions:**

NO

---

> ### Author Response · Authors · 2025-11-20
> **Official Response to Reviewer 9JFF - Comment #1 & #2**
>
> We thank the reviewer for the thoughtful feedback and for highlighting XIL as **“a more realistic and challenging benchmark for continual learning”** and XEED as **“a highly innovative solution”** to bidirectional knowledge transfer. We appreciate the constructive comments and will address all noted concerns in the response below. All clarifications and supplementary analyses from our response are included in the Discussion section of the paper’s Appendix.
>
> 1. Adaptation of Baseline Methods to the XIL Setting
>     - Thank you for raising this important point regarding how baseline methods were used in the multi-domain XIL setting. In general, adapting a baseline is necessary only when a method cannot be trained or evaluated in a new setting without altering its core architecture or training pipeline. However, **CIL and XIL share the same fundamental training pipeline**, and the key difference lies instead in the **underlying data-distribution assumptions** these methods implicitly rely on. Consequently, none of the baseline methods we evaluate encounter procedural incompatibilities when applied to XIL.
>     - CIL can be viewed as a special case of XIL in which all tasks—and both training and test phases—share a domain-invariant data distribution, a strict and often unrealistic assumption. XIL generalizes this setting by introducing domain variation, yielding a more realistic scenario where models must generalize to previously unseen distributions. Within this broader formulation, the data-distribution assumptions that CIL methods rely on are no longer satisfied, leading to a natural degradation in their performance. **Our objective is precisely to expose this distributional fragility.** Introducing additional XIL-specific “adaptation mechanisms” would alter these methods beyond their intended design, ultimately obscuring their inherent limitations under distribution shift.
>     - Importantly, we thoroughly examined a wide spectrum of CIL paradigms and selected the **strongest-performing and most widely adopted methods** as baselines to ensure a fair and competitive comparison. One might argue that, because XEED incorporates a generative model, generative-replay–based approaches would be the most appropriate baseline family. However, as shown in the table below, published results demonstrate that **prompt-based CIL methods consistently outperform generative replay on the same benchmark datasets.** Therefore, we compare XEED against **state-of-the-art prompt-based methods** rather than generative replay approaches, ensuring that XEED is compared against the strongest available CIL techniques. To make this clear, we provide a summary of published results on the most commonly used CIL benchmark, CIFAR-100:
>
>
>         | Method Type | Performance on 10-task CIFAR-100 |
>         | --- | --- |
>         | **Replay-based** [R1] | 58.40% |
>         | **Prompt-based** [R2] | 87.82% |
>
> 2. Ensuring Reproducibility for the XIL Benchmark
>     - We fully agree that ensuring reproducibility is essential, especially for a paper that introduces a new problem setting and a novel evaluation metric. **To support follow-up research, we have included the reference implementation code for XIL setting and the evaluation metric in the Appendix** (Sec. A.2 XIL Task Construction and Evaluation Code) **.** In addition, we are preparing a public GitHub repository that will contain the proposed method, all experimental code, and the additional experiments conducted during the rebuttal period. This will allow the community to easily verify our results, adopt the XIL benchmark, and build upon our work in a transparent and reproducible manner.

---

> ### Author Response · Authors · 2025-11-20
> **Official Response to Reviewer 9JFF - Comment #3**
>
> 3. Beyond Individual Components: XEED’s Contribution
>     - We acknowledge that the individual components of XEED, domain-specific prompts, generative replay, and prototype-based classifiers, are not entirely new in isolation. However, these components have never been integrated to address the challenges unique to the XIL setting. Existing work does not explain **how prompts should be designed to transfer domain knowledge across classes, how prototypes should be defined for unseen class–domain pairs, nor how both forward and backward domain transfer can be realized within a continual learning system.** XEED is the first approach to bring these capabilities together within a single, coherent architecture, establishing a new design direction that previous literature has not explored.
>     - Recent prompt-based approaches often increase the number of trainable parameters to better capture task-specific variations. While this strategy can improve in-domain performance, it frequently leads to **poor generalization to unseen domains**—a pattern we observed and empirically demonstrated in our experiments addressing **Reviewer RYCp’s question on “3. Adaptability Under Severe Domain Shifts.”** To examine how **the amount of trainable parameters** influences generalization to unseen domains, we implemented an adapter-based variant of XEED by inserting residual adapters after each ViT block. Despite containing substantially more trainable parameters, this adapter-augmented model fails to generalize to unseen class–domain combinations, exhibiting an **11% drop in F-BiDoT**. These findings indicate that simply adding parameter-efficient tuning modules does not address the fundamental challenge in XIL. The results on Office-31 are presented below:
>
>
>         | Method | F-BiDoT | A-BiDoT | Final | Avg |
>         | --- | --- | --- | --- | --- |
>         | **Original (Ours)** | **78.08** | **73.67** | **80.72** | **83.19** |
>         | **Adapter Variant** | 67.97 | 60.85 | 76.71 | 81.55 |
>     - Additionally, determining **how to construct prototypes for unseen class–domain combinations** is a challenge that prior continual learning methods have not addressed. **Pure generative replay is insufficient for transferring domain knowledge**, because replayed samples still intertwine class-specific and domain-specific factors. XEED overcomes this limitation by generating prototypes through **representation modulation**, explicitly disentangling domain attributes from class semantics. This disentanglement enables bidirectional transfer across unseen class–domain pairs. Without such separation, replayed samples cannot support domain-level generalization.
>     - Moreover, because existing CIL methods are built exclusively for **forward knowledge transfer** and typically rely on linear classifiers rather than prototype-based mechanisms, they inherently lack the ability to support **bidirectional (forward and backward) domain knowledge transfer**. Consequently, they offer no means for newly acquired domain information to propagate back to previously learned classes, leaving backward domain transfer entirely unaddressed.
>     - For these reasons, addressing XIL requires **mechanisms that go beyond what existing CIL methods support**, and XEED offers the first unified framework that enables consistent **bidirectional domain knowledge transfer across evolving class–domain combinations.**
>     - Our empirical findings indicate that relying solely on discriminative objectives (e.g., cross-entropy), as is common in recent continual learning research, **fails to robustly disentangle and transfer domain knowledge**. Discriminative models emphasize fine-grained, class-specific cues that sharpen decision boundaries, but they often overlook broader distributional structure. In contrast, generative models such as diffusion models learn to reconstruct or approximate the full data distribution, thereby capturing global, domain-level characteristics that purely discriminative training typically misses. For stable discriminative learning and reliable domain separation, **both types of information are essential**. To the best of our knowledge, **no existing method explicitly unifies discriminative modeling with diffusion-based distribution modeling**, making this an **important and promising direction for future research**. Building on this gap, we plan to develop a **discriminative–generative hybrid model** grounded in XEED’s core principles, enabling efficient forward and backward transfer of domain knowledge.
>
> References:
>
> [R1] DiffClass: Diffusion-Based Class Incremental Learning (ECCV 2024)
>
> [R2] Consistent Prompting for Rehearsal-Free Continual Learning (CVPR 2024)

---

> ### Author Response · Authors · 2025-11-21
> **A Note of Thanks**
>
> Thank you for the time and effort you put into reviewing our work.
> Your questions helped us strengthen the contribution and think more deeply about its significance.
> If anything else arises, please feel free to leave another comment.
> Wishing you a wonderful end of the year and a happy new year!

---

> ### Comment · Reviewer_9JFF · 2025-11-26
>
> Thank you for your clarifications. This paper is now stronger in my view, so I have increased my score.

---

### Official Review · Reviewer_RYCp · 2025-10-31

**Soundness:** 3
**Presentation:** 3
**Contribution:** 3
**Rating:** 6
**Confidence:** 3

**Summary:**

This paper introduces Cross-Expanding Incremental Learning (XIL), a new setting that extends class-incremental learning to handle data from distinct and evolving domains. XIL requires models to perform bidirectional domain transferability, integrating new classes into old domains and adapting old classes to new ones. To address this challenge, the authors propose XEED, a framework that combines domain-specialized prompts, residual-guided modulation, and evolving prototypes to expand class semantics across domains. They also introduce the BiDoT Score, a metric that measures how well models generalize across unseen class–domain combinations. Experiments on benchmark datasets with significant domain shifts show that XEED outperforms existing methods in both accuracy and BiDoT scores.

**Strengths:**

- The paper is clearly written and well-structured, allowing readers to easily follow the logical flow and understand the motivation and proposed method at both conceptual and technical levels.

- The empirical results convincingly demonstrate the effectiveness of the proposed approach across multiple standard benchmark datasets.

**Weaknesses:**

- The **quality and fidelity of synthetic exemplars** present a potential limitation. The proposed method assumes that diffusion-based generation can effectively represent unseen domain–class combinations. However, such generative processes may produce low-fidelity or stylistically inconsistent samples, which could introduce noise into the evolving prototype updates and distort the learned feature space. A more detailed analysis of the generation quality—either through visual inspection, quantitative metrics such as FID or LPIPS, or ablation studies—would help validate whether the synthetic exemplars truly enhance domain transfer rather than acting as noisy augmentations.

- Another concern lies in the **high computational cost and reliance on diffusion models**. The method depends on a pre-trained diffusion generator to synthesize cross-domain exemplars, which is computationally expensive due to multiple denoising steps, high GPU memory requirements, and significant I/O overhead. While diffusion models contribute to the quality of generated samples, their use raises questions about scalability and practicality in large-scale or real-time incremental learning scenarios. A discussion of the computational budget, inference latency, and trade-offs between efficiency and performance would strengthen the paper’s experimental transparency.

- Finally, the framework’s **dependence on frozen backbones and domain prompts** may restrict adaptability under severe domain shifts. The feature extractor remains fixed during prompt tuning, assuming pre-trained features are sufficiently general for new domains. In cases where the domain gap is large (e.g., transferring from natural images to sketches or depth maps), this assumption may not hold, leading to suboptimal adaptation even with domain-specific prompts. Allowing partial backbone fine-tuning or incorporating adaptive normalization could improve flexibility and robustness in such scenarios.

**Questions:**

- How do the authors ensure that the diffusion-generated exemplars are of sufficient visual and semantic quality to represent unseen domain–class combinations?

- Could noisy or low-fidelity exemplars negatively affect the evolving prototype updates, and if so, how is this mitigated?

- How scalable is XEED when applied to larger datasets or a greater number of incremental tasks and domains?

- Why was the backbone
𝑓
𝜙
f
ϕ
	​

 kept frozen during prompt tuning, and have the authors experimented with partial fine-tuning or adapter layers?

---

> ### Author Response · Authors · 2025-11-20
> **Official Response to Reviewer RYCp - Comment #1**
>
> We thank the reviewer for the thoughtful feedback and for highlighting the **clarity of our presentation** and the **strength of our empirical results**. We also appreciate the reviewer’s constructive comments and questions, and we will address all identified weaknesses and concerns carefully in our response. All clarifications and supplementary analyses from our response are included in the Discussion section of the paper’s Appendix.
>
> 1. Impact of Synthetic Exemplar Quality on Prototype Dynamics
>     - Thank you for raising the important concern that synthetic images may be low-fidelity or stylistically inconsistent, potentially distorting prototype representations. As shown in **Fig. 7**, our t-SNE visualization indicates that **generated samples occupy semantic regions similar to those of real samples**, suggesting that the generative process does not introduce fundamentally incompatible features. Nevertheless, we acknowledge that a small number of stylistically inconsistent samples may arise, and therefore we conducted an additional analysis to quantify whether such samples could meaningfully distort prototype learning. **Our findings are summarized below, and this analysis will be incorporated into Sec. 5.4 (Additional Experimental Analyses) of the revised manuscript.**
>         - Standard metrics such as FID and LPIPS are not suitable for evaluating stylistic inconsistency in our setting. These metrics are intended for model-to-model comparison and measure how one generator approximates a real distribution, rather than providing absolute indicators of sample quality. Moreover, the reviewer’s concern pertains to **intra-class deviation**, i.e., whether some samples fall off the class–domain manifold. Neither FID nor LPIPS captures such within-distribution anomalies: FID compares two global distributions and cannot detect local outliers, while LPIPS requires paired reference images, which are unavailable in our setting. Thus, these metrics cannot determine whether generated samples are off-manifold or capable of distorting prototypes. We therefore adopt a sample-level deviation measure based on the Mahalanobis distance.
>         - **To approximate potentially problematic samples, we analyze the distribution of generated images within each class–domain** using CLIP embeddings. For every class–domain, we compute the **Mahalanobis distance** for all generated samples, treat the bottom 5% (closest to the distribution center) as normal examples, and treat the top 5% (farthest) as outliers that may represent stylistic deviations or lower fidelity. **This procedure directly reflects the reviewer’s concern, as these distant samples capture the very deviations that might fall off the typical class–domain manifold.**
>         - To assess whether such outliers distort prototypes, we compare prototype–normal distances with prototype–outlier distances. If outliers distorted prototypes, they would be closer to the prototype than normal samples, effectively pulling the prototype toward them. However, we consistently observe the opposite: as shown in the table below, the domain-level results—computed by averaging across all classes—indicate that outliers remain farther from the prototype, with **85–100% of classes exhibiting this pattern.** Because prototype updates use a mean-based aggregation, higher-quality samples naturally exert greater influence, preventing atypical outliers from meaningfully affecting prototype evolution.
>
>
>             | Domain (Office-31) | Prototype–Normal distance | Prototype–Outlier distance | % Outlier Farther | Prototype-GT Outlier distance |
>             | --- | --- | --- | --- | --- |
>             | Amazon | 25.2344 | 37.7467 | 85.0% | 41.0796 |
>             | DSLR | 19.8796 | 29.8873 | 100.0% | 23.5270 |
>             | Webcam | 23.7143 | 37.4090 | 95.2% | 27.2508 |
>         - We further compare these distances with those observed in the real (ground-truth) dataset. The absolute magnitude of prototype–sample distances in synthetic data is comparable to the natural intra-class diversity present in real images (see ‘Prototype–Outlier’ and ‘Prototype–GT Outlier’ columns in the table above). In some domains (e.g., amazon), real outliers are even farther from the prototype than synthetic outliers. This demonstrates that the variation introduced by generated samples lies well within the normal range of natural variation and is therefore neither abnormal nor harmful.
>         - **In summary**, although a minority of generated samples may be less typical, they do not bias or distort prototypes. Synthetic intra-class variation remains comparable to that of real data, and outlier samples stay sufficiently distant from the prototype to avoid exerting harmful influence on prototype updates.

---

> ### Author Response · Authors · 2025-11-20
> **Official Response to Reviewer RYCp - Comment #2 & #3**
>
> 2. Computational Overhead
>     - We appreciate the reviewer’s thoughtful comment regarding the computational overhead of using a diffusion model. We fully acknowledge that diffusion models can be expensive due to their iterative denoising process; however, our design intentionally minimizes this cost while preserving the performance benefits. As noted in Section 5.1 (Training Details), we use only 50 denoising steps, resulting in approximately 12 seconds of generation time per class on a single A5000 GPU, and each class requires just 25–30 synthetic exemplars. This keeps the overall computational budget modest, and importantly, the diffusion model is used only during training, meaning it has **no impact on inference latency**. Given that our approach yields **up to a 31.41% improvement in F-BiDoT compared to the second-best baseline** (Table 1-PACS), we believe this is a reasonable and well-justified trade-off.
>     - Our empirical analysis further suggests that relying solely on discriminative objectives (e.g., cross-entropy), which are commonly adopted in recent continual learning research, **is insufficient for robustly disentangling and transferring domain knowledge**. Classification models trained with discriminative objectives tend to emphasize fine-grained, class-relevant cues that shape sharp decision boundaries, but often fail to capture broader distributional properties. In contrast, generative models such as diffusion models optimize for reconstructing or approximating the full data distribution, enabling them to capture global, domain-level characteristics that are typically overlooked by purely discriminative training. For stable discriminative learning and accurate domain separation, we argue that both types of information are essential. To the best of our knowledge, **no existing approach explicitly integrates discriminative models with diffusion-based distribution modeling within a unified architecture**, and exploring this direction represents **a promising avenue for future research.** Building on the core idea of XEED, we plan to develop a **lightweight discriminative–generative hybrid model** that enables bidirectional domain transfer with far lower computational cost, ultimately replacing diffusion models without compromising performance.
>
> 3. Adaptability Under Severe Domain Shifts
>     - We agree that introducing more trainable components such as adapters can improve performance on seen class–domain combinations. However, our primary focus in this work is **generalization to unseen class–domain pairs**, where overfitting to the training class–domain distribution can be detrimental. Allowing the backbone to adapt too flexibly risks absorbing the biases inherent to the seen distribution, leading to the well-known trade-off between in-domain fitting and out-of-domain generalization. For this reason, our method employs a **frozen backbone with lightweight domain prompts**, which intentionally constrains adaptation while preserving generalizability. Importantly, this design not only provides strong robustness on unseen class-domain pairs but also maintains **competitive performance on seen distribution**, demonstrating that small prompts are sufficient to maintain high in-domain accuracy.
>     - To further examine this point, we implemented an adapter-based variant of XEED by attaching residual adapters to the output of each block in the ViT backbone, and compared it with the prompt-based version proposed in our paper. The results on Office-31 are summarized below. This analysis will be included in Sec. 5.4 (Additional Experimental Analyses) of the revised manuscript.
>
>
>         | Method | F-BiDoT | A-BiDoT | Final | Avg |
>         | --- | --- | --- | --- | --- |
>         | **Original (Ours)** | **78.08** | **73.67** | **80.72** | **83.19** |
>         | **Adapter Variant** | 67.97 | 60.85 | 76.71 | 81.55 |
>     - The results show that the generalization gap reaches nearly 11% in F-BiDoT for the adapter variant, indicating a substantial degradation in unseen class–domain performance when increasing the number of trainable parameters. At the same time, our method achieves **higher Final and Avg scores**, showing that it preserves strong performance even on seen domains.

---

> ### Author Response · Authors · 2025-11-20
> **Official Response to Reviewer RYCp - Comment #4**
>
> 4. Response to Questions
>     - How do the authors ensure that the diffusion-generated exemplars are of sufficient visual and semantic quality to represent unseen domain–class combinations?
>         - As shown in **Fig. 7**, our t-SNE visualization indicates that diffusion-generated exemplars occupy semantic regions closely aligned with those of real samples, demonstrating strong **semantic consistency**. Moreover, **Fig. 6** provides qualitative evidence that the synthesized samples retain sufficient **visual fidelity** to represent unseen domain–class combinations.
>     - Could noisy or low-fidelity exemplars negatively affect the evolving prototype updates, and if so, how is this mitigated?
>         - Please refer to our detailed analysis in **1. Impact of Synthetic Exemplar Quality on Prototype Dynamics**, where we show that outliers remain sufficiently distant from prototypes and therefore do not distort prototype evolution.
>     - How scalable is XEED when applied to larger datasets or a greater number of incremental tasks and domains?
>         - The DomainNet dataset contains more than three times as many classes as CIFAR-100, which is commonly used in incremental learning.
>         - As shown in the table below, when we increase task granularity by splitting each incremental task in half (resulting in a total of 6 tasks, with 2 tasks per domain in Office-31), baseline methods experience severe degradation in generalization due to the reduced variety of class–domain combinations encountered during training. In contrast, XEED maintains strong generalization to unseen domain–class pairs even in these more demanding settings. XEED outperforms the class-incremental state-of-the-art method CPrompt, demonstrating XEED’s scalability and its robustness under settings with increased task granularity and domain variability. These results will be added to Sec. 5.4 (Additional Experimental Analyses) of the revised manuscript.
>
>
>             | Method | F-BiDoT | A-BiDoT | Final | Avg |
>             | --- | --- | --- | --- | --- |
>             | **XEED (Ours)** | **84.40** | **80.92** | **80.75** | **84.19** |
>             | CPrompt | 70.12 | 71.72 | 73.75 | 79.72 |
>     - Why was the backbone kept frozen during prompt tuning, and have the authors experimented with partial fine-tuning or adapter layers?
>         - Please refer to our response in **3. Adaptability Under Severe Domain Shifts**, where we discuss why using small trainable parameters is essential for preventing overfitting to training distribution and present empirical comparisons with an adapter-based variant.

---

> ### Author Response · Authors · 2025-11-28
> **A Note of Thanks**
>
> Thank you for the time and effort you put into reviewing our work. Your thoughtful requests for clarification and additional experiments have been very helpful in improving the manuscript and clarifying its contribution. We are genuinely sorry that ICLR has closed the discussion phase, especially since we have not yet had the chance to hear whether your remaining concerns have been fully resolved. We would have very much welcomed the opportunity to continue the exchange. If any additional comments come to mind, we would be very grateful if you could share them—either through the system, if further comments are still possible, or later on by email after the review process has ended. Wishing you a relaxing end of the year and a joyful start to the new one!

---

### Author Response · Authors · 2025-11-20
**Global Response**

We thank all the reviewers for the thoughtful and constructive feedback. We appreciate the recognition of XIL as **“a more realistic and challenging benchmark for continual learning”** and **“a new paradigm of continual learning,”** as well as the acknowledgement of XEED as **“a highly innovative solution”** with **“novel and sound”** mechanisms for **bidirectional knowledge transfer**. We are also grateful for the positive remarks regarding the **contribution of the BiDoT metric**, **the clarity of our presentation and motivation**, and the **strength of our empirical results**.

We value all reviewer’s comments and questions, and we have carefully addressed all identified weaknesses and concerns in our response. All clarifications and supplementary analyses are included in the Discussion section of the paper’s Appendix. Additionally, for points noted as being clarified in the main text, corresponding revisions have been made to improve the clarity and completeness of the manuscript.

---

### Meta-Review · Area_Chair_L7wj · 2026-01-05

**Summary:**

The paper introduces Cross-Expanding Incremental Learning (XIL), a novel and challenging continual learning setting that captures bidirectional class–domain expansion, and proposes XEED, a carefully designed framework to address this problem. Reviewers raised concerns regarding the overlap between XIL and existing domain-related continual learning settings, the perceived incremental nature of the method’s components, reliance on diffusion-based generation, and the fairness of baselines under a new evaluation protocol.

The authors provided clear and technically sound rebuttals, clarifying how XIL differs from prior settings, why existing baselines are fundamentally limited under this assumption, and why the proposed design choices are necessary for bidirectional domain transfer. Additional analyses addressing robustness, domain order sensitivity, and prompt selection behavior strengthened the submission. Multiple reviewers explicitly indicated that their concerns were resolved and increased their scores accordingly.

**Reviewer Concerns:**

**Concerns addressed by the rebuttal**:
1. Relationship between XIL and existing continual learning settings
The rebuttal clarified how XIL differs from cross-domain continual learning, domain adaptation, and domain generalization, particularly with respect to the absence of multi-domain supervision per class and the requirement for bidirectional class–domain transfer.

2. Baseline selection under the XIL setting
The authors explained why standard CIL and prompt-based methods were used as baselines and why methods relying on domain adaptation or multi-source supervision are not directly applicable under XIL.

3. Use of diffusion-generated exemplars
Additional analyses addressed concerns about whether synthetic samples introduce noise or distort prototype updates, and provided quantitative information about the computational cost of diffusion-based generation.

4. Method design details and robustness
The rebuttal included further experiments and clarifications regarding frozen backbones, prompt selection behavior, domain order sensitivity, and scalability.

5. Reproducibility and experimental transparency
Details on benchmark construction, evaluation protocols, and code availability were added, addressing earlier concerns about reproducibility.

**Concerns that remain outstanding**:
1. Dependence on large pre-trained generative models
The method relies on access to strong diffusion models for generating synthetic exemplars, which could restrict its practical applicability.

2. Extent of methodological novelty
Some reviewers’ concerns regarding the lack of fundamentally new learning primitives remain only partially addressed, as the proposed approach constitutes a combination of existing techniques including prompts, generative replay, and prototype-based classification without introducing novel methodological insights.

**Reviewer Scores:**

During the discussion phase, the majority of concerns were effectively addressed. Reviewer 9JFF, gcQm, w46G, and EJAk have subsequently agreed to raise their scores. All reviewers have expressed positive opinions.

---

### Decision · Program_Chairs · 2026-01-26

Accept (Poster)